# When should I search more: Adaptive Complex Query Optimization with Reinforcement Learning

## Abstract

Query optimization is a crucial component for the efficacy of Retrieval-Augmented Generation (RAG) systems. While reinforcement learning (RL)-based agentic and reasoning methods have recently emerged as a promising direction on query optimization, most existing approaches focus on the expansion and abstraction of a single query. However, complex user queries are prevalent in real-world scenarios, often requiring multiple parallel and sequential search strategies to handle disambiguation and decomposition. Directly applying RL to these complex cases introduces significant hurdles. Determining the optimal number of sub-queries and effectively re-ranking and merging retrieved documents vastly expands the search space and complicates reward design, frequently leading to training instability. To address these challenges, we propose a novel RL framework called Adaptive Complex Query Optimization (ACQO). Our framework is designed to adaptively determine when and how to expand the search process. It features two core components: an Adaptive Query Reformulation (AQR) module that dynamically decides when to decompose a query into multiple sub-queries, and a Rank-Score Fusion (RSF) module that ensures robust result aggregation and provides stable reward signals for the learning agent. To mitigate training instabilities, we adopt a Curriculum Reinforcement Learning (CRL) approach, which stabilizes the training process by progressively introducing more challenging queries through a two-stage strategy. Our comprehensive experiments demonstrate that ACQO achieves state-of-the-art performance on three complex query benchmarks, significantly outperforming established baselines. The framework also showcases improved computational efficiency and broad compatibility with different retrieval architectures, establishing it as a powerful and generalizable solution for next-generation RAG systems.

## 1 Introduction

Retrieval-Augmented Generation (RAG) has become a core paradigm in the LLM era because it grounds generation in external evidence, thereby improving factuality, recency, and attribution (Huang & Huang, 2024; Lewis et al., 2020). Achieving these benefits in RAG hinges on obtaining high-quality retrieved evidence, which in turn depends on transforming a user's natural-language question into a self-contained, retrieval-friendly query. This step is known as **Query Optimization (QO)** (Yu et al., 2020; Vakulenko et al., 2021; Zhang et al., 2024).

Existing QO techniques primarily optimize a single query through expansion or abstraction (Yu et al., 2020; Vakulenko et al., 2021; Zhang et al., 2024) in different approaches. Prompt-based approaches (Azad & Deepak, 2019) leverage meticulously crafted instructions to guide the LLM in generating more effective search queries. For instance, a simple prompt might instruct the LLM to "rephrase the user's question to be more suitable for a search engine.". Interactive-learning based methods (Xu et al., 2024; Zhu et al., 2025; Feng et al., 2023) go a step further by engaging in a feedback loop with the user or a simulated environment, allowing the model to refine its queries iteratively based on the quality of retrieved results. Pseudo-document generation techniques (Wang et al., 2023; Gao et al., 2023) transform the original query into a hypothetical, longer document that contains richer context, which can then be used to retrieve more relevant information from

the knowledge base. More recently, agentic and reasoning-augmented reinforcement learning (RL) methods—valued for their reduced dependence on labeled supervision—have shown strong empirical gains (Singh et al., 2025; Zhu et al., 2025). However, most of these solutions implicitly assume a one-to-one correspondence between a user query and an optimized query, which limits their coverage of complex information needs.

In real-world RAG applications, complex queries are common and often require multiple parallel or sequential sub-queries, notably for disambiguation and decomposition (Song & Zheng, 2024).

- **Disambiguation queries**, such as a user asking, "*When did Arsenal last win the FA Cup? [SEP] 2005 [SEP] What about them compared to Chelsea in league titles?*", require the system to interpret multi-turn contexts and clarify entity references (e.g., linking "them" back to Arsenal while introducing Chelsea for comparison). This may necessitate generating multiple parallel or sequential sub-queries to retrieve and contrast evidence.

- **Decomposition queries**, such as a user asking, "*What were the global shipments of iPhones in 2022 and 2023, respectively?*", require breaking down a multi-objective problem into independent sub-queries (e.g., "*global iPhone shipments in 2022*" and "*global iPhone shipments in 2023*"), retrieving results for each, and then synthesizing a final answer.

While some prior work has explored these problems (Ammann et al., 2025; Perez et al., 2020; Liu et al., 2024), applying reinforcement learning to such complex scenarios still presents a series of challenges: (1) deciding query number and depth (when to stop, whether to branch, how to merge); (2) performing multi-path retrieval and document aggregation across heterogeneous retrievers (sparse, dense, hybrid) with consistent, robust signals; and (3) coping with expanded search spaces and sparse/delayed rewards, which destabilize training. We argue that an effective QO system for complex queries should satisfy two goals:

- **Adaptive query handling**: it should adaptively decide the number and depth of sub-queries and switch among disambiguation, decomposition and single-query expansion and abstraction.

- **Stability and integrability**: it should support an end-to-end pipeline (query reformulation → multi-retrieval → document re-ranking → answer generation), seamlessly integrate with sparse and dense retrieval backends, and incorporate stabilizing training mechanisms tailored to RL.

To meet these goals, in this paper we propose Adaptive Complex Query Optimization (ACQO), an RL framework that learns when and how to expand the search process and how to accumulate evidence robustly. First, we let LLM decide whether to trigger decomposition or disambiguation, producing a set of parallel or staged sub-queries based on query complexity and intent diversity. Then, we perform model-agnostic re-ranking and fusion by jointly exploiting rank positions and retrieval scores, enabling smooth integration with heterogeneous retrievers and providing stable intermediate signals for the RL agent. Finally, we introduce a Curriculum Reinforcement Learning (CRL) strategy with two stages: an initial phase for broad exploration over all samples to establish general policies, followed by a focused phase that emphasizes challenging cases. This curriculum mitigates reward sparsity and improves convergence stability across the spectrum of query complexities. In experiments, ACQO achieves state-of-the-art performance on widely used RAG benchmarks, including conversational query reformulation (TopiOCQA) (Adlakha et al., 2022) and multi-hop reasoning (HotpotQA) (Yang et al., 2018), with additional out-of-domain evaluation on MultiHop-RAG (Tang & Yang, 2024) demonstrating strong generalization capabilities. Notably, our lightweight components achieve performance comparable to approaches requiring specialized retrieval modifications or complex re-ranking architectures, while maintaining significantly lower computational overhead. Experimental results demonstrate substantial improvements over baseline methods in both quantitative metrics and qualitative analysis. The contributions of this work are as follows:

- We propose ACQO, which unifies adaptive multi-query decision-making with robust evidence fusion in an end-to-end RL framework for complex queries.

- We introduce a universal re-ranking mechanism to combine rank positions and retrieval scores in a model-agnostic manner, improving stability and transferability across heterogeneous retrievers.

- Through extensive experiments on benchmark datasets, we demonstrate that ACQO significantly outperforms existing methods while maintaining computational efficiency, establishing its superiority for complex query processing in RAG systems.

Table 1: Performance comparison on easy vs. hard query subsets across datasets and retrievers (%).

| Method | TopiOCQA (Recall@10) | | | | HotpotQA (MAP@10) | | | |
| | ANCE | | BM25 | | ANCE | | BM25 | |
| | Easy | Hard | Easy | Hard | Easy | Hard | Easy | Hard |
| Prompt-based | 59.4 | 52.6 | 34.3 | 45.1 | 36.8 | 31.2 | 50.1 | 40.5 |
| SFT | 56.2 | 54.8 | 33.1 | 38.7 | 44.7 | 33.5 | 45.2 | 43.7 |
| Vanilla RL | 63.9 | 54.8 | 58.5 | 61.2 | 42.3 | 38.2 | 50.4 | 46.0 |
| ACQO (ours) | **66.2** | **58.0** | **60.3** | **64.5** | **50.4** | **41.5** | **53.1** | **48.3** |

## 2 WHAT MAKES QUERIES COMPLEX IN REAL-WORLD RAG SCENARIOS?

In this section, we conduct a systematic analysis of query complexity patterns in real-world RAG benchmark. By examining the inherent characteristics of queries across different datasets, we identify the key challenges that motivate our ACQO framework design.

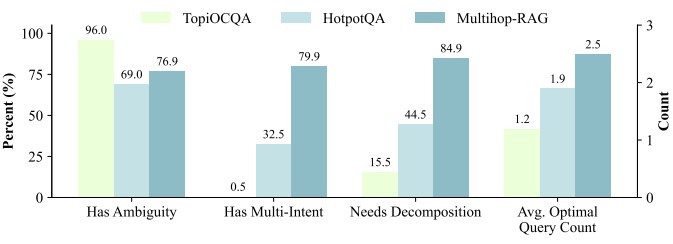

Figure 1: Query Complexity Distribution.

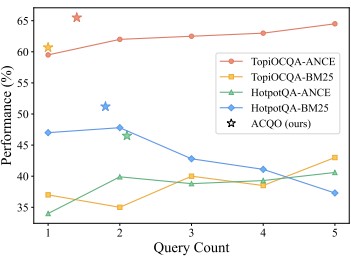

Figure 2: Performance metrics for different query counts.

### 2.1 QUERY COMPLEXITY ANALYSIS FRAMEWORK

We analyze three representative RAG benchmarks: TOPIOCQA for multi-turn conversational QA, HOTPOTQA for multi-hop factual reasoning, and MULTIHOP-RAG for real-world multi-hop retrieval. For each query, we conduct a structured analysis using the following criteria:

- **Ambiguity Detection**: Flag ambiguous entities or references that need disambiguation.
- **Multi-Intent Analysis**: Identify distinct intents embedded in the query.
- **Decomposition Assessment**: Judge whether decomposition improves answerability.
- **Optimal Granularity**: Identify the minimum number of sub-queries from the generated set that yields optimal retrieval metrics.

We analyze 200 representative queries from each dataset, focusing on understanding the distribution and characteristics of complex queries in real-world scenarios.

### 2.2 DATASET ANALYSIS: PREVALENCE OF COMPLEX QUERIES

Our structured analysis reveals significant complexity patterns across the three toy datasets, with Figure 1 illustrating the distribution of query characteristics. Specifically, a substantial proportion of queries are complex: on average, 48.3% require decomposition, and 37.6% exhibit multiple intents. Moreover, the optimal number of sub-queries varies across domains (1.2–2.5 on average), indicating that decomposition strategies must be context-sensitive rather than one-size-fits-all.

### 2.3 WHY CURRENT METHODS STRUGGLE WITH COMPLEX QUERIES

We evaluate representative query optimization approaches across different paradigms: prompt-based optimization using *DeepSeek-V3.1* (DeepSeek-AI, 2024) with decomposition prompts, supervised fine-tuning (SFT) via *Qwen2.5-3B* (Qwen, 2024) query rewriter, and vanilla reinforcement learning (REINFORCE with sparse rewards) also based on *Qwen2.5-3B*.

The performance analysis in Table 1 reveals critical limitations of existing approaches when handling complex queries. Current methods exhibit substantial performance variations between easy and hard queries, with SFT approaches showing dramatic drops of up to 11.2% on HotpotQA (44.7% to 33.5% with ANCE). Moreover, optimal approaches vary significantly across retrieval systems—vanilla RL excels with BM25 (61.2%) but degrades with ANCE (54.8%) on hard queries.Figure 2 further demonstrates that fixed decomposition strategies suffer from dual limitations in both efficiency and effectiveness. These inconsistent patterns highlight the absence of principled approaches for systematic query optimization, revealing three critical gaps: adaptive complexity recognition, retriever-aware optimization, and effective integration for decomposed queries.

## 3 ADAPTIVE COMPLEX QUERY OPTIMIZATION

### 3.1 TASK FORMULATION

In traditional Query Optimization (QO) pipeline, the task is defined as refining the query to retrieve the golden document(s) relevant to the user's current query and conversational context (if any) from a large collection of documents. Formally, given the current query $q^{(t)}$ ($t \geq 1$) and its historical context $C^{(t-1)} = \{(q_i, a_i)\}_{i=1}^{(t-1)}$ (if $t \geq 2$), where $t$ denotes the current turn number, a query optimization model $\Theta$ generates a de-contextualized query $\hat{q}^{(t)}$c. $\hat{q}$ (($t$) is omitted for simplicity) is subsequently input into a retrieval system, which returns a ranked list of the top-$k$ documents from the collection $\mathcal{P}$. We denote this ranked set as $\mathcal{R}_k(\hat{q}) = \{p_1, p_2, \ldots, p_k\}$, $\mathcal{R}_k(\hat{q}) \subseteq \mathcal{P}$, where $p_i$ represents the document ranked at position $i$. Let $\mathcal{P}^* \subseteq \mathcal{P}$ denote the set of golden documents corresponding to $\hat{q}$. The objective of QO is (1) to maximize the probability that at least one golden document in $\mathcal{P}^*$ appears in $\mathcal{R}_k(\hat{q})$; and (2) to minimize the ranking positions of the golden documents within $\mathcal{R}_k(\hat{q})$.

In our work, we extend this formulation by considering the disambiguation and decomposition scenarios, where an optimized query set $\hat{\mathcal{Q}}_q$ will be generated. Each sub-query $\hat{q}_q \in \hat{\mathcal{Q}}_q$ retrieves its own top-$k$ documents $\mathcal{R}_k(\hat{q}_q)$, and these candidates are subsequently combined and re-ranked to produce the final top-$k$ documents, denoted as $\mathcal{R}_k(\hat{\mathcal{Q}}_q)$. This design enhances both the coverage and ranking quality of golden documents.

### 3.2 OVERALL FRAMEWORK

As illustrated in Figure 3, ACQO proceeds in two curriculum reinforcement learning (CRL) stages: (1) *Explore CRL*, which promotes broad exploration and early stabilization; and (2) *Converge CRL*, which emphasizes precision and convergence on harder cases.

The core idea is to integrate query optimization with CRL in a fully self-directed manner. Without external supervision or intervention, the model learns to adaptively converge to suitable query numbers and optimization strategies across heterogeneous retrieval systems. In the following, we first introduce our re-ranker design, which consolidates multiple retrieval lists produced from the query set, and then detail the two-stage CRL procedure.

### 3.3 RE-RANKER DESIGN

**Method.** Inspired by Reciprocal Rank Fusion (RRF), we propose a new method named **Rank-Score Fusion (RSF)** to address two key limitations of RRF: it only considers rank positions while ignoring absolute retrieval scores, and it cannot properly handle cases where documents obtain identical ranks across multiple lists.

In RSF, each sub-query returns a ranked list of candidate documents, where each document is associated with a retrieval(e.g., ANCE) score and a rank position. For a given document $p$, we collect its appearances across all $M$ sub-queries into a set $\{(s_j, r_j)\}_{j=1}^{M}$, where $(s_j, r_j)$ denotes the score and rank of document $p$ in the $j$-th sub-query. We then compute two aggregated quantities for $p$:

$$P(p) = \frac{1}{\sum_{j=1}^{M} \frac{1}{r_j}}, \quad S(p) = \max_{j=1,\ldots,M} s_j. \tag{1}$$

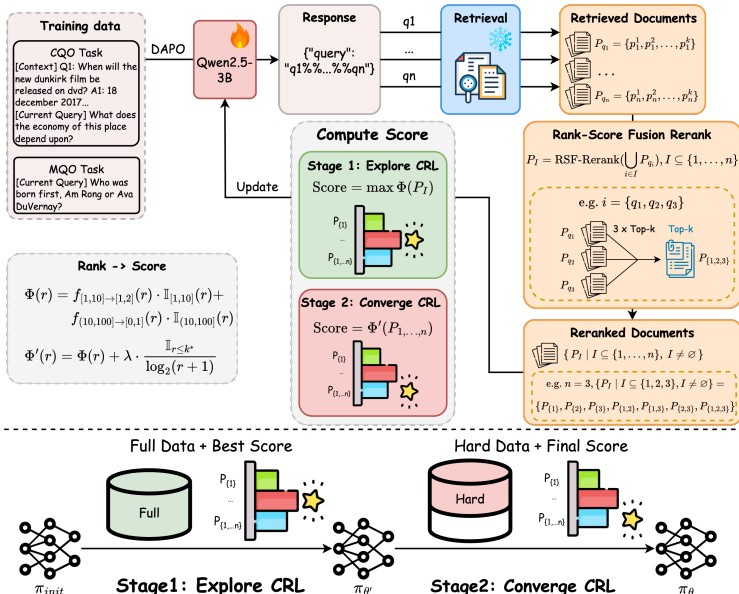

Figure 3: Overview of ACQO. ACQO employs two-stage curriculum reinforcement learning to adaptively optimize complex queries and integrate multi-retrieval results via Rank-Score Fusion.

Here, $P(p)$ reflects the combined influence of rank positions (relative values), while $S(p)$ captures the strongest absolute score observed for document $p$. We therefore perform lexicographical sorting with $P(p)$ as the primary key (ascending order: lower rank indicates better consensus) and $S(p)$ as the secondary key (descending order: higher score indicates stronger evidence). Formally, candidate documents are re-ranked according to:

$$\mathcal{R}_k = \text{Top-}k\big(\text{sort}\{(p, P(p), S(p)) \mid p \in \mathcal{R}_k(q_1) \cup \cdots \cup \mathcal{R}_k(q_M)\}\big), \tag{2}$$

where the sorting key is $(P(p), -S(p))$ in ascending order. This encodes a hierarchical preference: "Trust rank consensus first; use scores only to break ties among similarly-ranked documents."

**Advantages.** Our RSF method inherits RRF's simplicity and efficiency while extending its capability through score integration. RSF offers three key advantages: (1) **Zero latency overhead**: introduces no inference delay and seamlessly integrates with neural re-rankers. (2) **Universal compatibility**: directly applicable to both sparse (e.g., BM25) and dense (e.g., ANCE) retrievers across different index structures. (3) **Enhanced robustness**: leverages both rank positions and absolute scores for more balanced re-ranking while resolving rank ambiguities.

### 3.4 CURRICULUM REINFORCEMENT LEARNING (CRL)

#### 3.4.1 BASE REWARD FUNCTION

We build upon the Rank-Incentive Reward Shaping (RIRS) framework proposed in ConvSearch-R1 (Zhu et al., 2025), which provides dense rank-based reward signals and alleviates the sparsity of traditional metrics such as NDCG and MRR. Here, the rank $r$ is defined as the position assigned to a document in the re-ranked list $\mathcal{R}_k$ from our RSF module. The base rank-to-score mapping employs a continuous piecewise linear transformation:

$$\Phi(r) = f_{[1,10] \to [1,2]}(r) \cdot \mathbb{I}_{[1,10]}(r) + f_{(10,100] \to [0,1)}(r) \cdot \mathbb{I}_{(10,100]}(r), \tag{3}$$

where $f_{A \to B}$ represents a linear mapping function from interval $A$ to interval $B$, $\mathbb{I}_A(r)$ is the indicator function that equals 1 when $r \in A$ and 0 otherwise, and $r$ is the rank variable.

To accommodate multiple relevant documents, we employ a weighted aggregation score emphasizes the most promising retrieval results. Suppose the rank of $n$ retrieved relevant documents in ranked set $\mathcal{R}$ are $r_1, r_2, ..., r_n$ respectively, the $r_i$ score is defined as:

$$s(r_i) = \eta^i \cdot \Phi(r_i), \tag{4}$$

where $\eta$ is the decay coefficient. This generalization retains the dense reward structure of RIRS while providing additional flexibility to adapt the weighting scheme for different retrieval scenarios.

Taking the format correctness into the consideration, the complete reward score is defined as:

$$S(\mathcal{R}) = \sum_{i=1}^{n} s(r_i) \cdot \mathbb{I}_{format} + \delta \cdot (1 - \mathbb{I}_{format}) \tag{5}$$

where $\mathbb{I}_{format}$ serves as the format compliance gate, and $\delta < 0$ represents the non-compliance penalty coefficient.

### 3.4.2 STAGE I: EXPLORE-ORIENTED CRL

**Data Curriculum.** In the exploration stage, we employ the full training dataset without filtering. This ensures that the model is exposed to both easy and hard cases, providing sufficient diversity to stabilize early training and improve robustness. By leveraging the entire dataset, the model can better explore the space of optimization without being biased toward specific difficulty levels.

**Reward Design.** Building upon the base reward function, Stage I encourages exploration by reinforcing the *combination of the best-performed sub-queries*. Suppose $\hat{\mathcal{Q}}$ is the set of optimized sub-queries, and for any non-empty subset $\hat{\mathcal{Q}}'$ in the power set of $\hat{\mathcal{Q}}$, denoted as $\mathcal{P}(\hat{\mathcal{Q}})$, we compute its the stage-specific reward as:

$$G^{(I)}(\hat{\mathcal{Q}}) = \max_{\hat{\mathcal{Q}}' \in \mathcal{P}(\hat{\mathcal{Q}}) \setminus \varnothing} S(\mathcal{R}_k(\hat{\mathcal{Q}}')). \tag{6}$$

This design allows the model to explore diverse decomposition strategies and ensures that promising sub-queries are strongly reinforced, even in the early stage when the model is not yet stable.

### 3.4.3 STAGE II: CONVERGE-ORIENTED CRL

**Data Curriculum.** In the convergence stage, we refine the training distribution by focusing on the tougher cases. Rather than arbitrary filtering, we identify the optimal learning frontier by analyzing the performance distribution of Stage I models.

Formally, let $\mathcal{Q}_{train}$ denote the full training query set. We define the *learning complexity score* for each input query $q$ as:

$$\tau(q) = \frac{1}{K} \sum_{k=1}^{K} G^{(I)}(\hat{\mathcal{Q}}_q^{(k)}), \tag{7}$$

where $K$ denotes the number of rollouts. The convergence curriculum $\mathcal{Q}_{conv}$ is constructed by retaining samples within *optimal challenge zone*:

$$\mathcal{Q}_{conv} = \{q \in \mathcal{Q}_{train} : \tau(x_i) \leq \tau_{thres}\} \tag{8}$$

where $\tau_{thres}$ is the theoretical boundary indicating when retrieval performance is sufficiently complex to continue learning without destabilizing optimization.

This principled approach ensures that the model focuses on samples that are neither trivially easy (already mastered) nor prohibitively difficult (leading to sparse learning signals), thereby maximizing learning efficiency in the convergence phase.

**Reward Design.** Stage II transitions from exploratory reward maximization to precision-focused optimization via a reward architecture that emphasizes ranking quality over quantity exploration. The Stage II reward function directly evaluates the complete sub-query ensemble:

$$G^{(II)}(\hat{\mathcal{Q}}) = S(\mathcal{R}_k(\hat{\mathcal{Q}})). \tag{9}$$

To address the inherent challenge of sparse positive signals in top-ranked positions, we introduce a *logarithmic precision weighting* mechanism, inspired by NDCG's theoretical foundation, which reflects the information-theoretic principle that higher-ranked results contribute exponentially more to user satisfaction, which is defined as:

$$\Phi'(r) = \Phi(r) + \lambda \cdot \frac{\mathbb{I}_{r \leq k^*}}{\log_2(r + 1)}, \tag{10}$$

where $\lambda > 0$ is a precision amplification parameter, and $k^*$ represents the critical ranking threshold, $\mathbb{I}_{r \leq k^*}$ is the indicator function ensuring bonuses apply only to top-tier results.

This bonus-based design provides stronger incentives for exact top placements while still leveraging the smooth decay of $\Phi(r)$ for other positions. As a result, the model gradually shifts from broad exploration in Stage I to precise convergence in Stage II.

## 4 EXPERIMENTS

### 4.1 EXPERIMENTS SETUP

**Datasets.** We train and evaluate our model on three representative benchmarks that cover both **multi-turn** conversational query optimization, which primarily focus on query **disambiguation**, and **multi-hop** query optimization task focusing on query **decomposition**. For disambiguation task, we use TopiOCQA (Adlakha et al., 2022), a challenging open-domain conversational QA dataset with topic shifts. For decomposition task, we adopt HotpotQA (Yang et al., 2018) and evaluate generalization on MultiHop-RAG (Tang & Yang, 2024), a RAG-focused multi-hop retrieval benchmark.

**Baselines.** We compare against three categories of prior work. For single query optimization reformulation and abstarction, we include *IterCQR* (Jang et al., 2024), *ADACQR* (Lai et al., 2025), and *ConvSearch-R1* (Zhu et al., 2025). For query optimization with expansion, we evaluatedd *LLM4CS-RAR* (Mao et al., 2023), *CHIQ-Fusion* (Mo et al., 2024), *RETPO* (Yoon et al., 2025), and *AdaQR* (Zhang et al., 2024). For the complex query optimization setting, as there are no dedicated methods, we construct few-shot prompting baselines by adapting the above methods. We report post optimization retrieval performance after applying each baseline's optimization procedure.

The details regarding retriever, implementation and evaluation metrics are provided in Appendix A.

### 4.2 MAIN RESULTS

Table 2 and 3 show the retrieval performance of our method on TopiOCQA and HotpotQA using BM25 and ANCE retrievers, along with comparisons to baselines.

The results on TopiOCQA demostrate that ACQO significantly outperforms most methods across different retrieval settings. Notably, our method achieves competitive performance (34.9% MRR@3, 37.7% NDCG@3) using self-supervised via retrieval feedback, while *ConvSearch-R1* achieves a strong 37.8% MRR@3 in sparse retrieval, this performance stems primarily from its extended reasoning process and aggressive rewrite expansion mechanisms, which are also present in other methods. As shown in Table 8, its strong performance comes at the cost of over $10\times$ more tokens than our method, making it too slow and resource-heavy for practical end-to-end RAG use, which gains driven by scale, not scalable design. However, ACQO demonstrates superior generalization capabilities, achieving the best R@10 (62.6%) and R@100 (83.2%) performance on sparse retrieval. In dense retrieval settings, ACQO shows remarkable effectiveness, attaining competitive MRR@3 (36.6%), NDCG@3 (39.4%) and R@10 (65.6%), demonstrating its ability to work across different retrieval architectures.

On HotpotQA, using only a 3B parameter model, ACQO achieves the best results across all metrics under both sparse and dense retrieval settings. Notably, ACQO outperforms ConvSearch-R1 on this more challenging multi-hop dataset (49.6% vs. 44.4% MAP@10), demonstrating superior decomposition capability. Query decomposition generally helps models outperform their non-decomposition counterparts; yet even the strongest decomposition baselines (e.g., DeepSeek V3.1) fall short of the raw query baseline in sparse retrieval. This indicates that straightforward decomposition or instruction-based rewriting can harm retrieval effectiveness on this multi-hop dataset. In contrast, ACQO avoids such degradation and significantly outperforms the raw query: in sparse retrieval, it achieves 86.9% R@4 (+3.6%) and 91.6% R@10 (+2.7%); in dense retrieval, it reaches 82.2% R@4 (+13.9%) and 85.8% R@10 (+11.0%), outperforming the best baseline by +4.8% and +3.3% respectively. These results demonstrate that ACQO successfully bridges the gap between query decomposition and retrieval alignment, delivering superior and robust performance without relying on larger models or sacrificing efficiency.

Table 2: Retrieval performance comparison on TopiOCQA (%). **NS** denotes training without rewrite supervised data, and **NCoT** denotes training without chain-of-thought reasoning.

| Method | NS | NCoT | Sparse(BM25) | | | | Dense(ANCE) | | | |
|---|---|---|---|---|---|---|---|---|---|---|
| | | | MRR@3 | NDCG@3 | R@10 | R@100 | MRR@3 | NDCG@3 | R@10 | R@100 |
| DeepSeek-V3.1 | - | - | 15.5 | 17 | 36.7 | 65.3 | 28.4 | 30.8 | 56.3 | 77.8 |
| vanilla RL *(Qwen2.5-3B)* | - | - | 31.2 | 36.1 | 60.8 | 82.5 | 34.5 | 38.3 | 62.1 | 81.1 |
| IterCQR *(T5-base)* | ✗ | ✓ | 16.5 | 14.9 | 29.3 | 54.1 | 26.3 | 25.1 | 42.6 | 62.0 |
| ADACQR *(T5-base+LLaMA7B)* | ✗ | ✓ | 28.3 | 26.5 | 48.9 | 71.2 | 38.5 | 37.6 | 58.4 | 75.0 |
| LLM4CS-RAR *(ChatGPT)* | ✓ | ✗ | 27.9 | 26.4 | 48.4 | 71.1 | 35.4 | 34.4 | 55.2 | 72.2 |
| CHIQ-Fusion *(T5-base+LLaMA2-7B)* | ✗ | ✓ | 25.6 | 23.5 | 44.7 | – | 38.0 | 37.0 | 61.6 | – |
| RETPO *((LLaMA2-7B)* | ✗ | ✓ | 28.3 | 26.5 | 48.3 | 73.1 | 32.2 | 31.1 | 51.6 | 69.5 |
| AdaQR *(T5-base)* | ✗ | ✓ | 20.3 | 18.0 | 37.1 | 66.2 | 38.1 | 36.6 | 61.3 | 79.9 |
| ConvSearch-R1 *(Qwen2.5-3B)* | ✗ | ✗ | **37.8** | 36.2 | 59.6 | 80.1 | **50.5** | 50.1 | 72.0 | 86.3 |
| ACQO *(ours, Qwen2.5-3B)* | ✓ | ✓ | 34.9 | **37.7** | 62.6 | 83.2 | 36.6 | 39.4 | 65.6 | 85.1 |

Table 3: Retrieval performance comparison on HotpotQA (%).(qd: query decomposition)

| Type | Method | R@4 | R@10 | R@100 | MAP@10 | MRR@10 | NDCG@10 |
|---|---|---|---|---|---|---|---|
| Sparse (BM25) | Raw | 83.3 | 88.9 | 96.7 | 49.5 | 75.4 | 70.5 |
| | Qwen2.5-3B-inst (wo/qd) | 72.0 | 79.3 | 89.7 | 41.2 | 64.2 | 60.5 |
| | Qwen2.5-3B-inst (w/qd) | 75.3 | 81.2 | 89.5 | 42.7 | 65.9 | 62.4 |
| | DeepSeek-V3.1 (w/qd) | 81.1 | 86.6 | 93.3 | 49.1 | 70.6 | 66.2 |
| | vanilla RL *(Qwen2.5-3B)* | 82.3 | 89.9 | 95.6 | 48.8 | 77.5 | 73.2 |
| | ConvSearch-R1 *(Qwen2.5-3B)* | 83.0 | 90.2 | 96.0 | 51.1 | 77.0 | 72.3 |
| | ACQO *(ours, Qwen2.5-3B)* | **86.9** | **91.6** | **97.5** | **51.2** | **77.7** | **74.2** |
| Dense (ANCE) | Raw | 68.3 | 74.8 | 86.1 | 34.8 | 60.4 | 59.5 |
| | Qwen2.5-3B-inst (wo/qd) | 64.6 | 70.7 | 81.8 | 32.8 | 56.6 | 56.0 |
| | Qwen2.5-3B-inst (w/qd) | 67.0 | 73.0 | 81.8 | 34.9 | 57.5 | 57.3 |
| | DeepSeek-V3.1 (w/qd) | 77.4 | 82.5 | 88.9 | 46.1 | 66.8 | 65.7 |
| | vanilla RL *(Qwen2.5-3B)* | 79.2 | 83.9 | 89.1 | 41.1 | 75.5 | 74.4 |
| | ConvSearch-R1 *(Qwen2.5-3B)* | 75.0 | 79.4 | 87.5 | 44.4 | 72.8 | 72.2 |
| | ACQO *(ours, Qwen2.5-3B)* | **82.2** | **85.8** | **91.2** | **49.6** | **73.4** | **73.6** |

## 4.3 EVALUATION ON OUT-OF-DISTRIBUTION (OOD) DATA

A critical strength of our ACQO framework lies in its strong generalization to entirely unseen datasets. As shown in Table 4, when evaluated on MultiHop-RAG, ACQO consistently outperforms raw queries and all baselines across different retrievers. It achieves 49.7% R@4 compared to 45.7% for raw, with clear gains of 4% using *llm-embedder* and 3% using *bge-large-en-v1.5*, confirming its compatibility with varying retrieval architectures. ACQO maintains strong performance on unseen domains and query types, indicating it learns domain-invariant reformulation principles. All gains are achieved zero-shot without fine-tuning, which confirming it generalizes beyond dataset-specific patterns, making it highly adaptable to real-world retrieval systems with shifting data.

## 4.4 ABLATION STUDY

In this work, we have presented ACQO with three core components: Query Decomposition (QD) for adaptive query optimization, Rank-Score Fusion (RSF) for robust result aggregation, and a two-stage Curriculum Reinforcement Learning approach for stable training. We conduct comprehensive ablation studies on these components across both TopiOCQA and HotpotQA datasets to understand their individual contributions. As shown in Table 5, all three components are essential for optimal performance, with removing any single component leading to noticeable performance drops across both dense and sparse retrievers.

**Rank-Score Fusion (RSF)** emerges as the most critical component, with its removal causing the most significant performance degradation on TopiOCQA (37.7% → 35.0% NDCG@3 for sparse, 39.4% → 38.8% for dense), demonstrating that effective aggregation of multiple query results is fundamental to our approach. **Curriculum Reinforcement Learning** shows dramatic impact on training stability, with substantial performance drops without it (37.7% → 24.9% NDCG@3 for sparse on TopiOCQA), indicating that the convergence phase is essential for stable learning. We argue that Stage I (exploration) discovers diverse query reformulation strategies, while Stage II (convergence) refines these strategies for optimal performance. **Query Decomposition (QD)** shows moderate but consistent improvements (37.7% → 36.5% NDCG@3 for sparse on TopiOCQA), which aligns with expectations since TopiOCQA primarily involves disambiguation rather than complex query decomposition, yet QD still provides benefits for handling multi-faceted information needs.

Table 4: Retrieval performance comparison on MultiHop-RAG (%).

| Method | bge-large-en-v1.5 | | | | llm-embedder | | | |
|---|---|---|---|---|---|---|---|---|
| | MRR@10 | MAP@10 | R@10 | R@4 | MRR@10 | MAP@10 | R@10 | R@4 |
| Raw | 45.5 | 21.5 | 81.3 | 62.5 | 32.9 | 14.4 | 65.7 | 45.7 |
| Qwen2.5-3B(w/qd) | 44.8 | 21.2 | 80.5 | 61.7 | 33.2 | 14.7 | 65.7 | 45.9 |
| ACQO *(ours)* | **47.7** | **23.6** | **84.0** | **65.5** | **35.6** | **17.3** | **72.6** | **49.7** |

Table 5: Ablation study on retrieval performance (%).

| Dataset | TopiOCQA | | | | | | HotpotQA | | | | | |
|---|---|---|---|---|---|---|---|---|---|---|---|---|
| Retriver | Sparse | | | Dense | | | Sparse | | | Dense | | |
| Method | NDCG@3 | R@3 | R@10 | NDCG@3 | R@3 | R@10 | MAP@10 | R@3 | R@10 | MAP@10 | R@3 | R@10 |
| - wo/ RSF | 35.0 | 42.1 | 58.8 | 38.8 | 46.6 | 63.4 | 51.2 | 83.5 | 91.1 | 49.0 | 80.1 | **85.6** |
| - wo/ Stage II | 24.9 | 30.6 | 49.1 | 36.3 | 44.2 | 64.9 | **52.0** | 84.6 | **91.8** | 40.6 | 69.4 | 75.1 |
| - wo/ QD | 36.5 | 44.1 | 61.1 | 38.7 | 46.1 | 63.2 | 49.9 | 83.1 | 90.5 | 42.3 | 79.6 | 84.7 |
| **ACQO** | **37.7** | **45.8** | **62.6** | **39.4** | **47.8** | **65.6** | 51.2 | **84.8** | 91.6 | **49.4** | **80.5** | 85.6 |

The synergistic effects of all components create a robust framework where each component compensates for the limitations of others, establishing that ACQO requires all three components working in concert to achieve state-of-the-art performance.

## 4.5 TRAINING DYNAMICS ANALYSIS

Figure 4 illustrates the training progression of our two-stage curriculum learning approach on TopiOCQA and HotpotQA datasets. The results demonstrate the expected behavior of our adaptive query optimization framework.

As shown in both datasets, the average query count follows a characteristic *explore-then-converge* pattern: initially increasing during Stage I (exploration) as the model learns to decompose complex queries, then stabilizing or slightly decreasing during Stage II (convergence) as the model refines its decomposition strategies. This behavior aligns with our curriculum learning design, where the model first explores diverse query reformulation patterns before converging to optimal strategies.

The retrieval performance (R@10 for TopiOCQA, MAP@10 for HotpotQA) shows consistent improvement throughout training, with merged subqueries significantly outperforming baselines and approaching the performance of best subqueries. Notably, different retrievers result in different optimal query counts after training, which corroborates our finding that effective query optimization requires retriever-specific adaptation.

The training dynamics validate that our two-stage approach successfully balances exploration and exploitation, achieving both improved retrieval effectiveness and computational efficiency through adaptive query count optimization. We demonstrate the effectiveness of ACQO through state-of-the-art performance on TopiOCQA and HotpotQA datasets. In addition, the experimental results indicate that ACQO learns retriever-specific optimization strategies, with different retrievers yielding different optimal query patterns. Furthermore, ACQO exhibits superior performance in challenging settings such as generalization on unseen datasets and computational efficiency with smaller models.

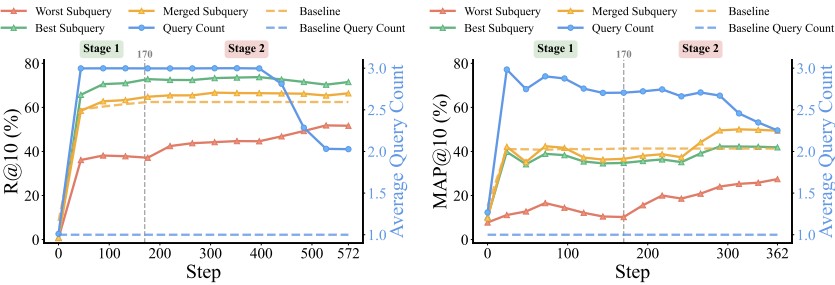

Figure 4: Query Adaptation and Performance Improvement on TopiOCQA(L) and HotpotQA(R).

Table 6: Efficiency Analysis: Inference Latency and Training Cost

(a) Avg Inference Latency (ms,TopiOCQA-ANCE)

| Method | #Q | Gen | Retri | Rerank | Tot | Speed |
|---|---|---|---|---|---|---|
| SFT (Qwen2.5-3B) | 2514 | 297 | 27 | 0 | 324 | 1.09× |
| ACQO | 2514 | 320 | 30 | 5 | **355** | 1.0× |
| ConvSearch-R1 | 2514 | 3230 | 25 | 0 | 3255 | **0.11×** |

*9.16× faster than ConvSearch-R1; +31ms for +8.2% MRR@3 vs. SFT*

(b) Training Cost (HotpotQA-ANCE)

| Method | GPU-H | Conv | MAP@10 |
|---|---|---|---|
| Vanilla RL | 8.4 | No | 41.1 |
| SFT + RL | 15.4 | yes | 45.3 |
| **ACQO (Stage I)** | 4.2 | yes | 42.3 |
| **ACQO (Full)** | 12.1 | yes | 49.6 |

## 4.6 LATENCY AND COST ANALYSIS

**Inference Latency Analysis.** Table 6a presents a detailed breakdown of inference latency across different pipeline stages. Our measurements on TopiOCQA-ANCE with a single H20 GPU show that ACQO adds only 31ms overhead compared to the SFT baseline (355ms vs. 324ms). More importantly, ACQO is 9.16× faster than ConvSearch-R1 (355ms vs. 3255ms) while maintaining comparable accuracy (as shown in Tables 2 and 3). This substantial speedup makes ACQO a Pareto-optimal choice for production deployment, offering the best balance between accuracy and efficiency.

The latency breakdown reveals that the additional overhead primarily comes from query generation (+23ms) and retrieval (+3ms), with our lightweight Rank-Score Fusion module contributing only 5ms. This validates our design philosophy of achieving strong performance through algorithmic innovations rather than computationally expensive components.

**Training Cost Analysis.** Table 6b compares training costs on HotpotQA-ANCE using 8 H20 GPUs. Full ACQO training requires 12.1 GPU-hours, comparable to the SFT+RL baseline (15.4 GPU-hours) but without requiring any supervised query rewriting data. Notably, ACQO-Stage I converges in only 4.2 GPU-hours while achieving 42.3% MAP@10, demonstrating efficient initial exploration.

While vanilla RL appears faster (8.4 GPU-hours), it fails to converge properly, getting stuck at a low performance ceiling (41.1% MAP@10) due to training instability. The root cause is insufficient valid samples—the DAPO algorithm fails to collect enough qualified samples within its sampling budget (`max_num_gen_batches=20`), causing premature termination with suboptimal performance. This validates the necessity of our curriculum learning strategy for stable convergence.

These results demonstrate that ACQO achieves superior performance with practical computational costs: (1) Inference efficiency: 9.16× faster than ConvSearch-R1 with minimal overhead over SFT; (2) Training efficiency: comparable cost to SFT+RL but without supervised data requirements; (3) Training stability: successful convergence where vanilla RL fails. This favorable efficiency-accuracy trade-off establishes ACQO as a practical solution for production RAG systems.

## 5 CONCLUSION

In this work, we propose ACQO, a two-stage reinforcement learning framework that addresses complex query optimization in RAG systems through self-supervised retrieval feedback, which leverages retrieval signals via adaptive query decomposition and rank-score fusion to provide retriever-specific guidance for query optimization. Experimental results demonstrate state-of-the-art performance on TopiOCQA and HotpotQA, while achieving 9.1× faster inference than strong baselines. Our analysis further reveals that ACQO learns retriever-specific optimization strategies, with each retriever yielding distinct optimal query patterns. Furthermore, our framework demonstrates superior performance in challenging scenarios, including strong generalization to unseen datasets and efficient operation with smaller models, establishing a powerful, efficient, and generalizable solution for next-generation RAG systems.

ETHICS STATEMENT

Our work complies with the ICLR Code of Ethics. The datasets used in this study are publicly available benchmark datasets and do not contain any personally identifiable or sensitive information. No human subjects, private user data, or personally identifiable information were involved in data collection or model training. All experiments were conducted using open-source frameworks and standard computational resources. We acknowledge that large-scale machine learning models may potentially amplify biases present in the training data. To mitigate this risk, we carefully followed the dataset usage guidelines and report all evaluation details transparently to encourage reproducibility and further scrutiny by the community.

REPRODUCIBILITY STATEMENT

We have made every effort to ensure the reproducibility of our results. All datasets used in this work are publicly available benchmark datasets. To facilitate replication, we provide detailed specifications of the retriever implementations, training parameters, prompts employed, evaluation metrics, and all related parameter settings in the main paper and the appendix. In the supplementary material, we also provide the core code used in this paper, including scripts for launching the retrieval service, training, and evaluation, all with comprehensive comments for clarity.

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

## THE USE OF LARGE LANGUAGE MODELS (LLMS)

In this work, large language models were used solely as auxiliary tools to support writing and editing, such as grammar checking and formatting suggestions. No part of the core scientific contribution, including the design of methods, experiments, or analyses, relied on outputs generated by LLMs. All technical content, results, and conclusions were conceived and verified entirely by the authors. The authors bear full responsibility for any content generated with the assistance of LLMs.

## A EXPERIMENTAL DETAILS

### A.1 RETRIEVAL

In TopiOCQA and HotpotQA, we use the BM25 retriever implemented by Pyserini (Lin et al., 2021), and the ANCE retriever implemented by Faiss (Johnson et al., 2019). The hyperparameters of BM25 are set to $k_1 = 0.9, b = 0.4$ for TopiOCQA, and $k_1 = 1.2, b = 0.75$ for HotpotQA during all training and evaluation. For ANCE, to improve training efficiency, we first generate embeddings for documents and then build an HNSW index using Faiss's `IndexHNSWFlat`, with parameters $M = 64$ and *ef_construction* $= 2000$. The index construction for HotpotQA partially follows (Jiang et al., 2025). During evaluation, we use `IndexFlatIP` to construct a flat index to ensure accuracy. In MultiHop-RAG, we follow its original setup with the LlamaIndex (Liu, 2022) framework and adopt BGE-large-en-v1.5 and LLM-Embedder as retrievers. Both retrievers use a chunking strategy with `chunk_size`=256 and `chunk_overlap`=20, splitting the 609 original documents into 7786 chunks. For BGE-large-en-v1.5, we follow the official recommendation and add the instruction "Represent this sentence for searching relevant documents:" when converting text into embeddings.

## A.2 TRAINING AND EVALUATION

**Evaluation Metrics.** For TopiOCQA, we employ Mean Reciprocal Rank@K (MRR@K), Normalized Discounted Cumulative Gain@K (NDCG@K), and Recall@K (R@K) as evaluation metrics. For HotpotQA and MultiHop-RAG, we additionally use Mean Average Precision@10 (MAP@10) for assessment. For MultiHop-RAG, we follow the evaluation code and metric provided in the benchmark.

**Retrieval Systems.** We evaluated the performance of model under both sparse and dense retrievers. For TopiOCQA and HotpotQA, we select **BM25** as the sparse retriever and ANCE as the dense retriever, where **ANCE** (Xiong et al., 2020) is trained on MS-MARCO (Bajaj et al., 2016) document retrieval tasks. For MultiHop-RAG, we use **bge-large-en-v1.5** (Xiao et al., 2024) and **llm-embedder** (Zhang et al., 2023) as the retrievers.

**Implementation.** We deploy Qwen2.5-3B as the backbone and train the model individually on TopiOCQA and HotpotQA, following the two-stage CRL described in §3. We use verl (Sheng et al., 2025b) as our RL training framework, and adopt DAPO (Yu et al., 2025) as the optimization algorithm, training the models under BM25 and ANCE retrievers independently.

**Training Hyperparameters.** We adopt the default hyperparameters established by ConvSearch-R1 (Zhu et al., 2025) and the verl framework (Sheng et al., 2025a), rather than performing dataset-specific tuning. The only modification we make is setting the maximum response length to 256 tokens (vs. 1024 in ConvSearch-R1), since ACQO generates concise sub-queries rather than chain-of-thought reasoning, reducing the required output length.

Specifically, for both TopiOCQA and HotpotQA, the models are trained under BM25 and ANCE retrievers with essentially the same hyperparameter configuration across both stages. We adopt DAPO optimization with mini-batch size 64 and micro-batch size 8 per GPU (8 GPUs in total). The actor learning rate is $1 \times 10^{-6}$ with gradient clipping at 1.0. Entropy regularization is disabled (entropy_coeff $= 0$). KL control is not used in reward shaping (use_kl_in_reward=False). The clipping ratios are set as $[0.2, 0.28]$ with an additional coefficient clip_ratio_c $= 10.0$, following the default configuration of DAPO. We sample $K = 8$ rollouts per query. Generation uses temperature 0.8, top-$p = 0.8$, and top-$k = -1$ during training; for validation we set temperature $= 0.7$, top-$p = 0.8$, and top-$k = 20$. Dynamic batch sizing is enabled for efficiency, with maximum batched tokens set to 11408 and GPU memory utilization capped at 0.8. We set the training batch size to 256 and the generation batch size to 512. For HotpotQA, the maximum prompt length is 512 tokens, while for TopiOCQA it is 1536 tokens. In both datasets, the maximum response length is fixed to 256 tokens. We set the decay coefficient $\eta = 0.6$ for HotpotQA, while $\eta = 1.0$ is used for TopiOCQA. For stage II reward design, we set $k^* = 3$ for TopiOCQA and $k^* = 0$ for HotpotQA. For the training epochs, Stage I CRL is trained for 2 epochs on TopiOCQA and 3 epochs on HotpotQA. Stage II CRL is trained for 10 epochs on TopiOCQA with ANCE retriever and 8 epochs on TopiOCQA with BM25 retriever. For HotpotQA, Stage II is trained for 4 epochs with ANCE and 6 epochs with BM25.

## A.3 DATASETS

We use three datasets in our experiments: TopiOCQA, HotpotQA, and MultiHop-RAG. All experiments are conducted on standard training/test splits and document collections, as summarized in Table 7. For HotpotQA, we follow the corpus provided by the BEIR benchmark (Thakur et al.), which standardizes the document collection for retrieval-based evaluation.

**Data Collection** Our method does not require any supervised data; instead, it employ RL with different levels of difficulty across the two RL stages(Section 3.4). In Stage I CRL, we use the full official training set for TopiOCQA. For HotpotQA, however, given the large training set size and relatively high initial performance, we first filter out the higher-performing samples and retain only 50% of the data for Stage I training. In Stage II CRL, we apply dynamic filtering with the Stage I model (Section 3.4.3), again retaining roughly 50% of the samples. Basically, we set $\tau_{thres} = \frac{5}{3}$ and rollouts $n = 8$. This guides the model to focus on moderately difficult instances, thereby improving learning efficiency and convergence in Stage II.

Table 7: Statistics of datasets used in our experiments. "#Golden / query" denotes the number of golden documents associated with each query.

| Dataset | Split | #Queries | #documents | #Golden / query |
|---|---|---|---|---|
| TopiOCQA | train
test | 45450
2514 | 25700592 | 1 |
| HotpotQA | train
test | 85000
7405 | 5233329 | 2 |
| MultiHop-RAG | test | 2556 | 7786 | multiple |

# B FURTHER EXPERIMENTS

## B.1 CASE STUDY

In this section, we begin with a representative case to further discuss how our method improves retrieval performance.

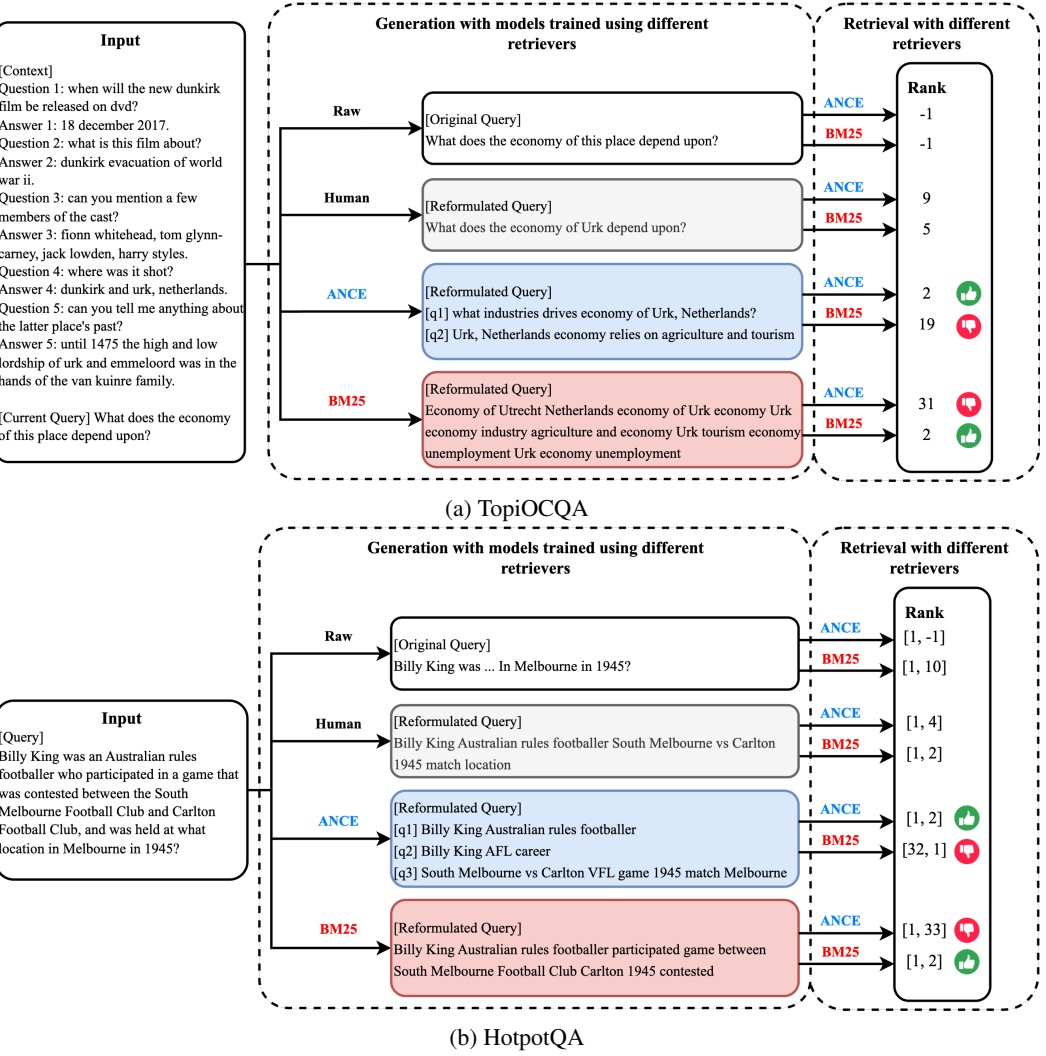

(a) TopiOCQA

(b) HotpotQA

Figure 5: Comparison of queries generated by models trained with different retrievers and their retrieval performance across different retrievers.

In Figure 5, we compare the performance across different training–retrieval combinations on different datasets, i.e., the effectiveness of reformulated queries generated by models trained with a specific retriever when evaluated on other types of retrievers.

From the perspective of retrieval performance, we observe that retrieval effectiveness drops significantly when switching retrievers; moreover, model-generated reformulations outperform human-written ones (i.e., reformulations that are intuitively considered correct). This suggests that the evaluation of query quality should be retriever-dependent and may not necessarily align with human intuition.

From the perspective of query generation behavior, queries generated with different retrievers vary in both quantity and style, indicating that the reformulation style learned by the model is closely tied to the retriever used during training.

**What behavior does the model learn when trained with a specific retriever?**  Models trained with the ANCE retriever tend to generate multiple queries resembling natural language questions or statements, capturing complete semantic relations and emphasizing keywords or core entities with fewer stopwords. In contrast, models trained with the BM25 retriever are inclined to generate a single query that explicitly enumerates all relevant keywords.

**Are these behaviors aligned with retriever preferences?**  Indeed, the observed behaviors are consistent with the characteristics favored by different retrievers. For dense retrievers such as ANCE, queries expressed in a natural language style, often decomposed into multiple sub-queries, better capture semantic relations and leverage embedding-based similarity. In contrast, sparse retrievers like BM25 prefer a single query containing exhaustive keyword coverage, where term frequency and exact lexical overlap dominate ranking. This alignment indicates that our model effectively adapts its reformulation strategy to the underlying retriever, learning to generate query styles that are inherently compatible with the retriever's scoring mechanism.

**Why does our method also yield improvements on TopiOCQA?**  TopiOCQA consists of single-intent questions, each associated with only one golden document, which suggests that the optimal query should ideally be a single reformulation. Traditional approaches mainly rely on **expansion**, where the model generates a lengthy reformulation that conveys complete semantic information while leveraging its parametric knowledge to give an answer of the question, thereby increasing semantic similarity with candidate passages (see Table 8). In practice, however, we observe that the model often employs **rephrasing**—for example, expressing the same intent as either a question or a declarative statement—to broaden the search space and consequently achieve better retrieval results. Its advantages lie in stronger readability and higher efficiency, while also mitigating the negative impact of erroneous expansions when the model encounters unfamiliar or ambiguous queries, thereby improving robustness.

Table 8: Comparison of generation lengths across methods.

| Method | Retrieval | Reasoning | Response | Total |
|---|---|---|---|---|
| ConvSearch-R1 | - | 106 | 248 | 354 |
| ACQO *(ours)* | ANCE | - | 28 | 28 |
| | BM25 | - | 36 | 36 |

**Why is human-preferred query reformulation worse than model-generated?**  Here we collectively refer to strong instruction models (e.g., DeepSeek-V3) and human-written rewrites as *human-preferred query reformulation*, since such models can generate queries that are generally regarded as high-quality under instruction or few-shot settings. In contrast, we denote reformulations produced by our trained models as *model-generated query reformulation*. However, experimental results show that human-preferred reformulations still underperform compared to our method or other advanced baselines. Based on the above analysis, we summarize two main reasons:

(1) **Human-preferred methods do not know what constitutes a retrieval-effective query.** They tend to follow human instructions by completing the current query with context or decomposing

multi-intent queries into several sub-queries. Yet, when no clear sub-intent is present, they fail to decide how to **decompose**, and typically do not perform **rephrasing** or **expansion**.

(2) **Human-preferred methods are not capable of generating retrieval-specific reformulations.** For example, their relative performance gap to state-of-the-art baselines is larger on BM25 than on ANCE, since—as we have observed earlier—some queries with "poorer readability" may actually perform better under BM25.

This finding suggests that analyzing retriever-specific data generated through ACQO can provide insights into retriever preferences, which in turn can be used to optimize prompts for large language models and improve their performance on query optimization tasks.

Taken together, our method enables the model, without any supervised data and solely using retrieval performance as the reward, to autonomously adapt to the retriever type. In doing so, the model is able to capture reformulation patterns that are more compatible with the retriever, ultimately leading to optimal reformulations.

## B.2 SCALING CAPABILITIES

Figure 6 presents the experimental results of our method with Qwen2.5-3B and Qwen2.5-7B. Across both datasets and retrievers, the larger model consistently achieves better performance, demonstrating that our approach exhibits strong scaling capability.

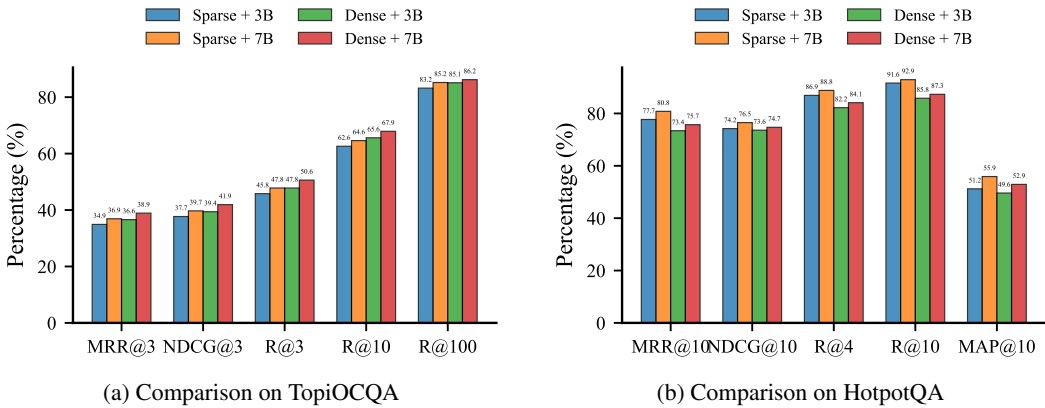

(a) Comparison on TopiOCQA      (b) Comparison on HotpotQA

Figure 6: Scaling Capabilities.

## B.3 SFT VS. RL COMPARISON.

Table 9 presents a systematic comparison of training strategies on the TopiOCQA dataset with the ANCE retriever. For SFT baselines, the training data are constructed by rolling out the Stage I CRL model under our framework and filtering out queries with poor rankings. These results collectively demonstrate that ACQO's two-stage curriculum reinforcement learning effectively addresses the fundamental challenges of complex query optimization, consistently outperforming both supervised baselines and vanilla RL approaches while maintaining training stability and data efficiency.

## B.4 END-TO-END QUESTION ANSWERING EVALUATION

While the retrieval metrics presented above demonstrate ACQO's effectiveness in query optimization, a critical question remains: do these retrieval improvements translate to better final answers in real-world RAG applications? To address this concern, we conduct comprehensive end-to-end question answering experiments that evaluate the complete RAG pipeline from query optimization to answer generation.

**Experimental Setup.** We use Qwen2.5-7B-Instruct (Qwen, 2024) as the reader model and DeepSeek-R1 (DeepSeek-AI, 2024) as the evaluation judge to assess answer quality on HotpotQA

Table 9: Comparison between SFT and RL methods on the TopiOCQA dataset with the ANCE retriever.

| Method | MRR@3 | NDCG@3 | R@3 | R@10 | R@100 |
|---|---|---|---|---|---|
| SFT | 28.4 | 30.7 | 37.3 | 53.4 | 71.5 |
| Vanilla RL | 34.5 | 38.3 | 34.6 | 62.1 | 81.1 |
| SFT + RL | 33.4 | 37.8 | 45.7 | 61.6 | 82.2 |
| Stage I only | 33.6 | 36.6 | 44.2 | 64.9 | **85.8** |
| Stage I + SFT | 28.5 | 30.8 | 37.7 | 53.3 | 70.2 |
| ACQO *(ours)* | **36.6** | **39.4** | **47.8** | **65.6** | 85.1 |

with the ANCE retriever. For each method (Raw Query, ConvSearch-R1, ACQO), we retrieve the top-10 documents using the optimized queries and provide them as context to the reader model, then evaluate the generated answers based on accuracy.

**Results and Analysis.** Table 10 presents the end-to-end evaluation results, comparing retrieval performance (MAP@10) with answer accuracy ($ACC_L$). The results reveal a strong correlation between retrieval quality and final answer accuracy across all methods. Starting from the raw query baseline (34.8% MAP@10, 16.4% $ACC_L$), ConvSearch-R1 achieves substantial improvements (44.4% MAP@10, 27.7% $ACC_L$), while ACQO further advances the state-of-the-art to 49.6% MAP@10 and 31.6% $ACC_L$.

Table 10: End-to-end question answering evaluation on HotpotQA-ANCE. MAP@10 measures retrieval quality, while $ACC_L$ evaluates final answer accuracy judged by DeepSeek-R1.

| Method | MAP@10 | $ACC_L$ | $\Delta ACC_L$ |
|---|---|---|---|
| Raw Query | 34.8% | 16.4% | - |
| ConvSearch-R1 | 44.4% | 27.7% | +11.3% |
| ACQO *(ours)* | **49.6%** | **31.6%** | **+15.2%** |

Notably, ACQO achieves a +3.9% improvement in $ACC_L$ over ConvSearch-R1, confirming that our curriculum reinforcement learning design effectively addresses the convergence challenges in mixed-complexity query optimization. This validates that the adaptive query decomposition and robust rank-score fusion mechanism not only improve retrieval metrics but also enhance the quality of final generated answers. Moreover, ACQO reaches 9.1× lower latency, representing a favorable efficiency-accuracy trade-off for production deployment.

## C  PROMPTS

Figure 7a shows the prompt used in ACQO, which remains the same across retrievers and datasets. If no context is available, it is set to empty. The same prompt is also employed in other experiments (e.g., ablation studies and supervised fine-tuning) with query decomposition. We also provide in Figure 7b the prompt version without query decomposition, which is used in experiments without query decomposition.

**Prompt**

You are an expert in generating queries for retrieval. Your task is to understand the intention of the user current question based on the historical conversation Q&A content (if any), and rewrite the question in a complete form.

- You need to retain the original query while expanding it with additional semantically relevant information. If no useful expansion is needed, return the original query as is.

- Your target is to output a rewrite to help search engines retrieve relevant documents effectively.

- You can generate multiple queries (no more than 3) if you think it's helpful for retrieval.

- You need to put your answer in JSON format, and name the key "query", for example: {"query": "xxx"}

- If you generate multiple queries, split them by %%, such as {"query": "xxx%%xxx%%xxx"}

## Historical Conversation Q&A Content

*{Context}*

## User Current Question

*{Question}*

(a) Prompt for standard ACQO

**Prompt without query decomposition**

You are an expert in generating queries for retrieval. Your task is to understand the intention of the user current question based on the historical conversation Q&A content (if any), and rewrite the question in a complete form.

- You need to retain the original query while expanding it with additional semantically relevant information. If no useful expansion is needed, return the original query as is.

- Your target is to output a rewrite to help search engines retrieve relevant documents effectively.

- You need to put your answer in JSON format, and name the key "query", for example: {"query": "xxx"}

## Historical Conversation Q&A Content

*{Context}*

## User Current Question

*{Question}*

(b) Prompt without query decomposition

Figure 7: Prompts used in our experiments.

