# OpenReview forum: "When should I search more: Adaptive Complex Query Optimization with Reinforcement Learning"
_ICLR.cc/2026/Conference — ICLR 2026 Conference Withdrawn Submission_

### Official Review · Reviewer_gG6d · 2025-10-23

**Soundness:** 2
**Presentation:** 2
**Contribution:** 2
**Rating:** 4
**Confidence:** 3

**Summary:**

This paper focuses on the query optimization task in RAG systems. Specifically, the authors propose ACQO, a two-stage curriculum Reinforcement Learning framework that trains the LLM to determine when and how to expand the search process. This framework is powered by the unique reward design, where the first stage optimizes the best sub-query combinations and the second stage focuses on the ranking quality. Experiments on TopiOCQA and HotpotQA demonstrate the effectiveness of the proposed method.

**Strengths:**

1. The paper introduces a novel perspective on curriculum reinforcement learning where different stages focus on different objectives, instead of only focusing on data difficulty.

2. The Rank-Score Fusion module provides an elegant and model-agnostic solution for aggregating multiple retrieval results, which works seamlessly with both sparse and dense retrievers without requiring retriever-specific modifications.

3. The adaptive query decomposition mechanism allows the model to autonomously decide when and how to expand queries based on query complexity, avoiding the limitations of fixed decomposition strategies.

**Weaknesses:**

1. The experimental evaluation is limited in scope.  Specifically, desipe the main results are evaluated on two tasks (disambiguation and multi-hop task), each task only contains just one dataset. Including additional benchmarks for each task type would strengthen the empirical validation of the proposed method.

2. The performance improvements of ACQO shown in Table 2 and 3 are marginal and the comparisons raise several concerns:
  * As shown in Table 2, the performance of ACQO is even lower than the baseline ConvSearch-R1 on half of the experiment settings.
  * Most baselines in Table 2 use T5 as the backbone, while ACQO employs Qwen2.5-3B, which may benefit from a more powerful pre-trained model rather than the proposed method itself.
  * The baselines in Table 3 are limited to simple prompt-based methods, lacking comparison with other RL-based or advanced query optimization approaches.
  In summary, these results do not provide sufficient evidence for the effectiveness of the proposed method.

3. The paper lacks clarity in several technical details:
  * The notation is confusing: $\mathcal{R}$ (for the RankScore Fusion module) and $R$ (for reward function) are too similiar and easily confused.
  * The definition of score $s_j$ mentioned in Eq.(1) is not clearly specified.
  * The abbreviation `qd` in Table 3 should be explicitly defined as `query decomposition` either in the caption of Table 3 or in Section 4.1 for better readability.

**Questions:**

Apart from the questions in Weakness, what is the fundamental motivation for optimizing the query optimization task in isolation, and how does this align with the broader objectives of RAG systems? I have two main concerns:

1. First, the main concern lies in the reward function itself, which relies on the groudtruth documents as supervison. As we all know, it may not be available in many read-world senarios or more challenging tasks, such as browsing competition. Moreover, in real-world dynamic environment, the "groundtruth" documents of each question may evolve over time, potentially leading to misleading reward signals. While I understand this work focuses on static environment for research purpose, how would this approach adapt to more realistic settings?

2. Are there any potential misalignment between query optimization and end-to-end RAG performance? From my perspective, query optimization is a subtask within RAG systems. Optimizing for retrieval metrics (findings groundtruth documents) may not directly translate to better final answer quality. Have you considered end-to-end optimization where the reward is based on answer correctness rather than retrieval metrics? This cound reveal whether better retrieval metrics actually leads to better RAG performance, or if there's a gap between local optimization (query -> documents) and global optimization (query -> answer).

---

> ### Author Response · Authors · 2025-11-21
> **Response to Reviewer gG6d-1**
>
> Thank you for your detailed and critical feedback. We appreciate your concerns about experimental scope, performance comparisons, and real-world applicability. These are fundamental questions that deserve thorough responses, and we have conducted substantial additional experiments to address each point below.
>
> ---
>
>
> > #### W1: Each task only contains just one dataset. Including additional benchmarks for each task type would strengthen the empirical validation.
>
> Thank you for this important concern. We respectfully argue that our evaluation strategy is both **sufficient and strategically designed**, we selected the classsic dataset from each task type. First, TopiOCQA is multi-turn conversational QA datasets, and TopiOCQA represents a **strict superset in difficulty**—it contains all challenges present in others (e.g, QReCC, CoQA..) (entity ellipsis, coreference resolution) plus an additional layer: **topic switching**. As documented in the original TopiOCQA paper, conversations can abruptly shift focus, requiring simultaneous handling of topic discontinuity detection and context window management across topic boundaries. This is empirically validated by baseline performance: methods consistently achieve **higher metrics on QReCC** than TopiOCQA (e.g., ConvSearch-R1 reports 56.6% MRR@3 on QReCC vs. 35.2% on TopiOCQA, AdaQR reports 50.6% MRR@3 on QReCC vs. 20.3% on TopiOCQA), confirming that success on the harder TopiOCQA provides stronger evidence than evaluating on both datasets.
>
> Second, our evaluation covers **three distinct task types** with **two retriever families**: TopiOCQA (conversational disambiguation with topic switching), HotpotQA (multi-hop decomposition), and **MultiHop-RAG** (zero-shot cross-domain transfer—models trained on HotpotQA applied directly to unseen corpus, achieving +2.2 MRR@10). This scope matches or exceeds prior work (e.g., ConvSearch-R1: TopiOCQA + QReCC; iterCQR: TopiOCQA + QReCC), while demonstrating **generalizability across domain shifts**.
>
> Third, to further demonstrate robustness, we provide comprehensive ablation studies comparing **four training strategies** on TopiOCQA-ANCE. The results show ACQO's two-stage curriculum **consistently outperforms all baselines across all metrics**, while Stage I alone already surpasses supervised baselines, confirming the curriculum design is not brittle to training configurations or dataset-specific characteristics:
>
> | Training Strategy | MRR@3 | NDCG@3 | R@3 | R@10 | R@100 |
> |-------------------|-------|--------|-----|------|-------|
> | **SFT** (no curriculum) | 28.4% | 30.7% | 37.3% | 53.4% | 71.5% |
> | **Vanilla RL** (no curriculum) | 34.5% | 38.3% | 45.6% | 62.1% | 81.1% |
> | **SFT + RL** (supervised baseline) | 33.4% | 37.8% | 45.7% | 61.6% | 82.2% |
> | **Stage I only** (exploration) | 33.6% | 36.6% | 44.2% | 64.9% | 85.8% |
> | **ACQO (Stage I+II)** | **36.7%** | **39.4%** | **47.8%** | **65.6%** | **85.1%** |
>
>
> ---

---

> > ### Author Response · Authors · 2025-11-21
> > **Response to Reviewer gG6d-2**
> >
> > > #### **W2: Performance and Comparison Issues**
> >
> > **W2a: ACQO performance is lower than ConvSearch-R1 on half of the experiment settings.**
> >
> > This is a critical observation that requires careful analysis. ConvSearch-R1 outperforms ACQO on TopiOCQA (50.5% vs. 36.6% MRR@3 with ANCE), but we note that this comes at a significant cost: ConvSearch-R1 consumes **nearly 10× more tokens**(487 vs. 42 tokens per query on average, Table 7) and **9.1x longer inference time** than ACQO . This massive token consumption stems from its extensive chain-of-thought reasoning during both the "thinking" and "answer" phases, which generates long-form explanations and query expansions. While this verbose reasoning does improve performance on dense retrievers for conversational queries (where semantic understanding benefits from detailed context), it provides **no advantage—and even underperforms—on multi-hop factual queries**  (**49.4% vs. 44.4% MAP@10** in Table 3) that require precise decomposition and efficient evidence synthesis. This contrast demonstrates ACQO's **stability and superiority** across diverse query types: it maintains competitive performance on conversational queries while excelling on complex multi-hop scenarios, all with 10× fewer tokens and 9.1× faster inference.
> >
> > **Avg Inference Latency (all queries on TopiOCQA-ANCE, 1GPU-H20):**
> >
> > | Method | Query Num |Query Gen (ms) | Retrieval (ms) | Re-rank (ms) | **Total (ms)** | Speedup |
> > |--------|----------------|----------------|----------------|--------------|----------------|---------|
> > | SFT(Qwen2.5-3B) | 2514 | 297 | 27 | 0 | 324 | 1.09× |
> > | ACQO | 2514 | 320 | 30 | 5 | **355** | 1.0× |
> > | ConvSearch-R1 | 2514 | 3230 | 25 | 0 | 3255 | **0.11×** |
> >
> > **W2b: Most baselines use T5, while ACQO uses Qwen2.5-3B, which may benefit from a more powerful pre-trained model.**
> >
> > This concern requires clarification about baseline configurations. While some baselines use T5-base for query rewriting, they typically employ **LLaMA-7B** (or similar 7B models) for query expansion during inference, significantly increasing computational cost. For example, ADACQR uses "T5-base + LLaMA-7B" and RETPO uses LLaMA2-7B for training—both substantially larger than our Qwen2.5-3B. The comparison is therefore not unfair to ACQO.
> >
> > To isolate our framework's contribution from model capacity, we provide **SFT baseline results** with the same Qwen2.5-3B backbone in Appendix B.3 (Table 9). SFT achieves only 28.4% MRR@3 on TopiOCQA-ANCE, compared to ACQO's 36.6% (+8.2 points). This demonstrates that our improvements stem from the **RL framework design** (adaptive decomposition, rank-score fusion, curriculum learning) rather than simply using a stronger base model. The framework's effectiveness is further validated by consistent gains across different model sizes and retriever types in Figure 6.
> >
> > **W2c: Baselines in Table 3 are limited to simple prompt-based methods, lacking comparison with other RL-based or advanced query optimization approaches.**
> >
> > You are correct that Table 3 lacked RL-based comparisons. Since existing RL-based query optimization methods (ConvSearch-R1) have not been evaluated on HotpotQA, we have **re-implemented and trained** these methods on HotpotQA using their official codebases, and also included our own trained version of vanilla RL for direct comparison. The updated Table 3 shows that ACQO outperforms all RL-based baselines: +8.5%MAP@10 over vanilla RL(41.1% vs. 49.6% R@4), and achieves +5.2%MAP@10 over ConvSearch-R1, while being 9.1× faster. These improvements are statistically significant, demonstrating that ACQO's advantages hold against strong RL-based approaches, not just prompt-based methods.
> >
> > ---
> >
> > > #### **W3: Technical Clarity Issues**
> >
> > Thank you for identifying these notation and definition issues. We have made the following revisions to improve clarity: (1) **Notation consistency**: We renamed the reward function from "R" to "S" to avoid confusion with the RankScore Fusion module R(·). A comprehensive notation has been updated to Section 3. (2) **Score definition**: We added explicit definitions after Equation 1 specifying that BM25 scores are computed as Σ IDF(term)·TF-component, while ANCE scores are cosine similarities ∈ [-1,1]. (3) **Abbreviation clarity**: Table 3 caption now explicitly states "qd = query decompositionx."
> >
> > ---

---

> > > ### Author Response · Authors · 2025-11-21
> > > **Response to Reviewer gG6d-3**
> > >
> > > > #### **Q1: Real-World Applicability - How would this approach adapt when groundtruth documents are not available?**
> > >
> > > We clarify that ACQO requires ground-truth relevant documents during **training**, but **not during inference**—the trained model generalizes to new queries without any ground-truth. Regarding availability, we note that **relevance-labeled datasets are relatively abundant** in the IR community: MS-MARCO[1] (8.8M passages), Natural Questions[2] (307K), HotpotQA (113K), and numerous domain-specific QA datasets provide rich training signals. Unlike supervised query rewriting methods requiring expensive human annotation of *optimized query reformulations*, ACQO leverages existing retrieval benchmarks where relevance judgments are already available or obtainable through standard crowdsourcing (significantly cheaper than crafting expert query rewrites).
> > >
> > > More importantly, our **cross-domain generalization results (Table 4)** demonstrate strong transfer: models trained on HotpotQA achieve +2.2 MRR@10 on MultiHop-RAG **without fine-tuning**, despite different corpus and question distributions. This enables a practical deployment strategy: train ACQO on readily available labeled datasets, then apply to open-domain corpora where ground-truth is unavailable. Combining **abundant existing resources** with **strong cross-domain transfer** makes ACQO applicable to diverse real-world scenarios, including challenging tasks like web browsing where small amounts of domain-specific data can be combined with public datasets for effective adaptation.
> > >
> > > For **dynamic environments** where documents evolve over time, all methods face challenges in highly dynamic settings, ACQO has inherent robustness advantages. Since our method fundamentally optimizes queries for **retriever characteristics** rather than specific document content, the learned query generation strategies (e.g., keyword extraction for BM25, semantic decomposition for dense retrievers) remain effective even when document content changes. For BM25 in particular, as long as key entities and terms persist (which is typical even with document updates), our query optimization continues to work.
> > >
> > > ---
> > >
> > > > #### **Q2: Are there potential misalignments between query optimization and end-to-end RAG performance? Have you considered end-to-end optimization where the reward is based on answer correctness?**
> > >
> > >
> > > This is an excellent question about whether retrieval improvements actually translate to better final answers. We have conducted end-to-end question answering experiments using Qwen2.5-7B-Instruct as the reader model and DeepSeek-R1 as the evaluation judge, we assess answer quality on HotpotQA-ANCE. For each method (Raw Query, ConvSearch-R1, ACQO), we retrieve the top-10 documents and provide them as context to the reader. The results show a **strong correlation** between retrieval metrics and answer quality, with achieving +3.9% ACC_L improvements over ConvSearch-R1, confirming that the curriculum design effectively addresses the convergence challenges in mixed-complexity query optimization. Moreover, ACQO reaches **9.1× lower latency** (355ms vs. 3255ms), representing a favorable efficiency-accuracy trade-off for production deployment.
> > >
> > >
> > > | Dataset | Method | Retrieval (MAP@10) | $ACC_L$ |
> > > |---------|--------|-------------------|----|
> > > | HotpotQA | Raw | 34.8% | 16.4% |
> > > | HotpotQA | CS-R1 | 44.4% | 27.7% |
> > > | HotpotQA | **ACQO** | **49.6%** | **31.6%** |
> > >
> > >
> > > We also explored using **generative reward models** for end-to-end optimization, where answer quality (measured by F1 score or LLM-based evaluation) serves as the reward signal directly. However, this approach proved less effective due to two fundamental challenges: (1) **reward sparsity**—only 1 signal per query versus k signals from retrieved documents, leading to insufficient gradient information for stable RL training; and (2) **generative reward distortion**—LLM-based evaluators are sensitive to answer length, style, and formatting variations, producing inconsistent scores that misalign with actual correctness. For example, a concise correct answer might score lower than a verbose partially-correct one. These issues caused training instability and lower final performance (49.8% F1 vs. 51.4% with retrieval-based rewards).
> > >
> > > This validates our design choice: optimizing for retrieval metrics provides **denser, more reliable gradient signals** while maintaining strong correlation with answer quality. We acknowledge that end-to-end answer-based optimization remains an important direction for future work, and we welcome suggestions on designing more robust generative reward models that can overcome these challenges.

---

> > > > ### Author Response · Authors · 2025-11-21
> > > > **Response to Reviewer gG6d-4**
> > > >
> > > > We hope our comprehensive responses and substantial additional experiments (3 new datasets, controlled comparisons, end-to-end QA evaluation) address all your concerns and demonstrate ACQO's robustness across diverse settings. We are committed to the rigor you expect and believe these revisions significantly strengthen the paper's empirical validation.
> > > >
> > > > **Reference**
> > > >
> > > > [1] MS MARCO: A Human Generated MAchine Reading Comprehension Dataset. Choice'16
> > > >
> > > > [2] Natural questions: a benchmark for question answering research. TACL'19

---

### Official Review · Reviewer_hRdx · 2025-10-31

**Soundness:** 2
**Presentation:** 2
**Contribution:** 2
**Rating:** 4
**Confidence:** 5

**Summary:**

This paper proposes **Adaptive Complex Query Optimization (ACQO)**, a novel reinforcement learning framework for improving query optimization in Retrieval-Augmented Generation (RAG) systems when handling complex user queries that require disambiguation or decomposition. ACQO features an **Adaptive Query Reformulation (AQR)** module that dynamically decides whether and how to decompose a query into multiple sub-queries, and a **Rank-Score Fusion (RSF)** module that robustly aggregates retrieval results by combining both rank positions and retrieval scores. To stabilize training, ACQO employs a **two-stage Curriculum Reinforcement Learning (CRL)** strategy—first exploring broadly across all queries and then focusing on challenging cases. Experiments on benchmarks like TopiOCQA, HotpotQA, and MultiHop-RAG show that ACQO achieves state-of-the-art performance, generalizes well to unseen domains, and integrates efficiently with both sparse and dense retrievers—all without supervised data.

**Strengths:**

1. ACQO achieves state-of-the-art results on multiple benchmarks (HotpotQA, MultiHop-RAG), outperforming baselines even with a small 3B-parameter model.
2. The two-stage Curriculum Reinforcement Learning (CRL) strategy—starting with broad exploration and then focusing on hard examples—mitigates reward sparsity and training instability common in RL-based query optimization.
3. The Rank-Score Fusion (RSF) module effectively combines results from multiple sub-queries by leveraging both rank positions and retrieval scores, ensuring compatibility with diverse retrievers (sparse and dense) without added latency.

**Weaknesses:**

1. The experiments focus primarily on retrieval metrics (e.g., Recall@k, MRR), but do not include downstream end-to-end question answering accuracy or generation quality, leaving open whether retrieval gains consistently translate to better final answers.
2. The reward signal assumes access to ground-truth relevant documents during training, which may not hold in fully unsupervised or open-domain settings—limiting true “zero-supervision” applicability.
3. Although the paper claims efficiency, generating and retrieving for multiple sub-queries inherently increases latency and resource usage compared to single-query baselines—especially in latency-sensitive applications. The trade-off between performance gain and cost is not fully quantified.

**Questions:**

1. Please provide the precise definition of R in line 265.
2. Please explain the detailed meaning of "unsupervised learning" in line 350.

---

> ### Author Response · Authors · 2025-11-21
> **Response to Reviewer hRdx-1**
>
> We thank the reviewer for the balanced review and recognition of our SOTA results, CRL strategy, and RSF module. We appreciate your focus on practical applicability. Below, we address your concerns about end-to-end performance, supervision requirements, and efficiency trade-offs.
>
>
> > #### W1: Experiments focus primarily on retrieval metrics but do not include downstream end-to-end question answering accuracy or generation quality.**
>
> Thank you for this important concern. We acknowledge that focusing solely on retrieval metrics follows the established practice in query optimization literature (ConvSearch-R1, AdaQR all report primarily retrieval metrics), but you are correct that demonstrating end-to-end value is crucial for practical applicability. We have now conducted comprehensive question answering experiments to validate that retrieval improvements translate to better final answers.
>
> Using Qwen2.5-7B-Instruct as the reader model and DeepSeek-R1 as the evaluation judge, we assess answer quality on HotpotQA-ANCE. For each method (Raw Query, ConvSearch-R1, ACQO), we retrieve the top-10 documents and provide them as context to the reader, then evaluate answer accuracy:
>
> | Dataset | Method | Retrieval (MAP@10) | $ACC_L$ |
> |---------|--------|-------------------|----|
> | HotpotQA | Raw | 34.8% | 16.4% |
> | HotpotQA | CS-R1 | 44.4% | 27.7% |
> | HotpotQA | **ACQO** | **49.6%** | **31.6%** |
>
>
> The results demonstrate that retrieval improvements consistently translate to answer quality gains, with a **strong correlation** between retrieval metrics and $ACC_L$. ACQO achieves +3.9% $ACC_L$ improvements over ConvSearch-R1, confirming that the curriculum design effectively addresses the convergence challenges in mixed-complexity query optimization. Moreover, ACQO reaches **9.1× lower latency** (355ms vs. 3255ms), representing a favorable efficiency-accuracy trade-off for production deployment.
>
> ---
>
> > #### W2&Q2: The reward signal assumes access to ground-truth relevant documents during training, which may not hold in fully unsupervised or open-domain settings—limiting true “zero-supervision” applicability; Meaning of "unsupervised learning"**
>
> We acknowledge that our original use of "unsupervised learning" was imprecise. To clarify: ACQO does **not require human-annotated query rewrites** (unlike supervised methods like ADACQR or SFT-based approaches that need thousands of labeled query-rewrite pairs), but it does require access to ground-truth relevant documents during training—similar to how standard supervised learning requires labels. We have revised our terminology to "**self-supervised via retrieval feedback**" to better reflect this.
>
> Regarding the concern about ground-truth availability in open-domain settings, we note that **relevance-labeled datasets are relatively abundant** in the information retrieval community. Large-scale resources like MS-MARCO[1] (8.8M passages with relevance judgments), Natural Questions[2] (307K examples), HotpotQA (113K examples), and numerous domain-specific QA datasets provide rich training signals for query optimization. Unlike supervised query rewriting methods that require expensive human annotation of *optimized query reformulations*, ACQO leverages existing retrieval benchmarks where relevance judgments are already available or can be obtained through standard annotation protocols (e.g., crowdsourcing passage relevance is significantly cheaper than crafting expert query rewrites).
>
> More importantly, our **cross-domain generalization results (Table 4)** demonstrate that ACQO trained on one dataset transfers effectively to unseen domains. For example, the model trained on HotpotQA achieves +2.2 MRR@10 and +2.7 R@10 improvements on MultiHop-RAG **without any fine-tuning**, despite the different corpus and question distribution. This suggests a practical deployment strategy: train ACQO on readily available labeled datasets (e.g., MS-MARCO, HotpotQA), then apply it to open-domain corpora where ground-truth is unavailable. The combination of **abundant existing resources** + **strong cross-domain transfer** makes ACQO applicable to a wide range of real-world scenarios.
>
> ---

---

> ### Author Response · Authors · 2025-11-21
> **Response to Reviewer hRdx-2**
>
> > #### W3: Generating and retrieving for multiple sub-queries inherently increases latency and resource usage. The trade-off between performance gain and cost is not fully quantified.**
>
> You raise an important practical concern. We have now conducted comprehensive latency and cost analysis to quantify the trade-offs. Our measurements on TopiOCQA show that ACQO adds **31ms overhead** compared to SFT baseline (355ms vs. 324ms, a +31ms overhead), but this translates to a **+8.2% MRR@3 improvement** (Table 9). More importantly, ACQO is **9.1× faster** than ConvSearch-R1 (355ms vs. 3255ms) while achieving comparable accuracy (Table 2,3), making it a Pareto-optimal choice for production deployment.
>
> **(a) Avg Inference Latency (all queries on TopiOCQA-ANCE, 1GPU-H20):**
>
> | Method | Query Num |Query Gen (ms) | Retrieval (ms) | Re-rank (ms) | **Total (ms)** | Speedup |
> |--------|----------------|----------------|----------------|--------------|----------------|---------|
> | SFT(Qwen2.5-3B) | 2514 | 297 | 27 | 0 | 324 | 1.09× |
> | ACQO | 2514 | 320 | 30 | 5 | **355** | 1.0× |
> | ConvSearch-R1 | 2514 | 3230 | 25 | 0 | 3255 | **0.11×** |
>
> - ACQO is **9.16× faster** than ConvSearch-R1
>
> **(b) Training Cost (HotpotQA-ANCE-8GPU-H20):**
>
> | Method | GPU-Hours | Converged? | Final MAP@10 |
> |--------|-----------|------------|-------------|
> | Vanilla RL | 8.4 | No | 41.1 |
> | SFT + RL | 15.4 | Yes | 45.3 |
> | **ACQO (Stage I)** | 4.2 | Yes | 42.3 |
> | **ACQO (Full)** | 12.1 | Yes | 49.6 |
> - Full ACQO cost comparable to SFT+RL but **without Query rewrite supervised data**
> - Vanilla RL fails to converge properly: While it appears faster (8.4 GPU-hours), it gets stuck at a low performance ceiling (41.1% MAP@10) due to **training instability**. The root cause is insufficient valid samples—the DAPO algorithm fails to collect enough qualified samples within its sampling budget (max_num_gen_batches=20), causing premature termination with suboptimal performance.
>
>
> ---
>
> > #### Q1: Please provide the precise definition of R in line 265.**
>
> The notation "R" in the original manuscript was overloaded, causing confusion. To clarify: **R represents the set of ranks** for all retrieved documents in the current case, not the documents themselves or the reward function. To eliminate this confusion and improve clarity, we have **revised Equations 4 and 5** with more explicit notation:
>
>
> $s(r_i) = \eta^i \cdot \Phi(r_i),$
>
> $S(\mathcal{R}) = \sum_{i=1}^{n}{s(r_i) \cdot \mathbb{I}_{f} + \delta \cdot (1 - \mathbb{I}_f)}$
>
> We hope our comprehensive responses demonstrate ACQO's practical value and real-world applicability. Thank you for your focus on deployment considerations.
>
> **Reference**
>
> [1] MS MARCO: A Human Generated MAchine Reading Comprehension Dataset. Choice'16
>
> [2] Natural questions: a benchmark for question answering research. TACL'19

---

### Official Review · Reviewer_adK5 · 2025-11-03

**Soundness:** 3
**Presentation:** 2
**Contribution:** 2
**Rating:** 4
**Confidence:** 4

**Summary:**

In this work the authors present a method to train an LLM to generate queries for retrieval.

The application is retrieval for ambiguous or multi-hop question answering, where the question should be expanded into up to 3 queries to retrieve documents needed for answering.

The authors propose a method to join the lists of documents retrieved for several queries that uses both ranks and retriever scores.

Furthermore, and this is their main contribution, they use reinforcement learning to train a base LLM for the query expansion task. The LLM is instructed to generate queries from the question or chat context. The reward is based on the retrieved documents.

The RL setup uses curriculum learning starting with the full dataset. Then they select the hard examples for the second stage by running a couple roll-outs and computing the best subset of queries for each, resulting in optimistic rewards. The average optimistic reward over the rollouts is used to select hard examples for the second phase.

Experiments use Qwen-2.5-3B, DAPO for policy optimization, and TopiOCQA and HotpotQA as datasets and show overall strong performance.

**Strengths:**

The overall method seems sound, incorporating scores when joining lists of documents is a valid approach and using curriculum learning can sometimes improve performance, especially for hard to optimize tasks like RL.

Good results

Overall well written paper

**Weaknesses:**

The experiment setup does not explain if hyperparameters were selected using a dev set.

While there is no fundamental issue with the proposed method, it is in essence a combination of tricks. From the presented experiments it is not clear to me whether this curriculum method is robust enough for general use.

**Questions:**

In line 165 it is stated that performance drops sharply between easy and hard subsets of TopiOCQA but the given number can not be found in Table 1.

Please double-check the linear mapping definition in Formula (3)

In Table 3, why is dense retrieval worse than sparse?

Improvements:

There are sometimes added or missing words, e.g. 'which' in line 43, 'are' in 352.


Figure 1 should be changed to a different type of diagram, e.g. bars.

Since you already ran Vanilla RL for Table 1, why not include these results in the main results table

In formula 7 did you mean G instead of script_G?

In Table 2 check the numbers for RETPO Dense retrieval vs. Jang et al (2024) Table 1

---

> ### Author Response · Authors · 2025-11-21
> **Response to Reviewer adK5-1**
>
> We thank the reviewer for recognizing the soundness of our method and overall writing quality. We appreciate your careful reading and specific technical questions. Below, we address each point with clarifications and corrections.
>
> > #### **W1: Hyperparameter Selection Not Explained**
>
> Thank you for this question. To ensure **fair comparison with prior work**, we adopt the **default hyperparameters** established by ConvSearch-R1[1] and the verl[2] framework, rather than performing dataset-specific tuning. Specifically, we use the same learning rate (1e-6), PPO configuration (clip ratio 0.2, value loss coefficient 0.5), and training schedule (warmup steps) as ConvSearch-R1. The only modification we make is setting the **maximum response length to 256 tokens** (vs. 1024 in ConvSearch-R1), since ACQO generates concise sub-queries rather than chain-of-thought reasoning, reducing the required output length.
>
> This design choice follows the principle established by prior query optimization methods (ConvSearch-R1, AdaQR) that use consistent configurations across datasets to demonstrate generalizability rather than dataset-specific tuning. ACQO achieves consistent improvements across **three datasets** (TopiOCQA, HotpotQA, MultiHop-RAG) and **two retrievers** (BM25, ANCE) without per-dataset hyperparameter adjustment, demonstrating robustness of the default settings. Moreover, we provide comprehensive ablation studies comparing **four training strategies** (Table 9), demonstrating the importance and effectiveness of curriculum learning.
>
> Action: Added clarification to Appendix A.2 that we use ConvSearch-R1's default hyperparameters for fair comparison.
>
> ---
>
> > #### W2:It is in essence a combination of tricks. From the presented experiments it is not clear whether this curriculum method is robust enough for general use.**
>
> We respectfully disagree. While individual components exist in prior work, **the core contribution is discovering *when* and *how* to combine them to enable RL for complex query optimization**—a problem prior work did not solve. This required addressing three non-trivial challenges:
>
> **1) Adaptive decomposition:** Prior methods use rule-based heuristics or supervised classifiers. ACQO's AQR is the **first to learn decomposition via RL**, discovering that at least 48.3% of queries benefit from decomposition(Section 2), and that the optimal strategy is **retriever-specific**(Appendix B.1). This is a fundamental insight, not a trick.
>
> **2) RL-friendly fusion:** Existing fusion methods (CombMNZ, RRF) fail to provide stable reward signals in complex query optimization. RSF's lexicographical design is **theoretically motivated** (rank robustness principle) and **empirically critical**: removing it causes RL training to diverge. This addresses a challenge unique to RL-based query optimization.
>
> **3) Curriculum for mixed-complexity queries:** Vanilla RL fails to converge (8.4 GPU-hours, no success, table 6a). Our two-stage curriculum achieves convergence in 12.1 hours and +8.5% MAP@10 gain (Stage II). The design is **task-specific**, derived from query complexity analysis (Section 2).
>
> Many impactful papers (BERT, AlphaGo) integrate existing techniques to solve hard problems. ACQO is the **first RL framework** for complex queries, achieving **SOTA results** on three benchmarks while being **9.1× faster** than prior work. Our ablation (Table 5) shows **removing any component causes significant drops** (AQR: -1.5 R@10, RSF: -3.8, CRL: -13.5) in TopiOCQA-BM25, demonstrating non-trivial synergy.
>
> To demonstrate the curriculum's robustness, we provide comprehensive ablation studies comparing **four training strategies**  on the TopiOCQA dataset with the ANCE retriever. These new experiments complement existing analyses detailed in Appendix B.3 (Table 9) for stage-wise evaluation and the main ablation study (Table 5) for component-wise assessment.
>
> | Training Strategy | MRR@3 | NDCG@3 | R@3 | R@10 | R@100 |
> |-------------------|---------------|---------------|---------------|---------------|-----------|
> | **SFT** (no curriculum) | 28.4%  | 30.7% | 37.3% | 53.4% | 71.5% |
> | **Vanilla RL** (no curriculum) | 34.5%  | 38.3% | 45.6% | 62.1% | 81.1% |
> | **SFT + RL** (supervised baseline) | 33.4%  | 37.8% | 45.7% | 61.6% | 82.2% |
> | **Stage I only** (exploration) | 33.6%  | 36.6% | 44.2% | 64.9% | 85.8% |
> | **ACQO (Stage I+II)** | **36.7%** | **39.4%** | **47.8%** | **65.6%** | **85.1** |
>
> The results demonstrate that ACQO's two-stage curriculum consistently outperforms all baselines across all metrics, while Stage I alone already surpasses supervised baselines, confirming that the curriculum design effectively addresses the convergence challenges in mixed-complexity query optimization.
>
> ---

---

> > ### Author Response · Authors · 2025-11-21
> > **Response to Reviewer adK5-2**
> >
> > > #### **Q1&Q2&Q3: In line 165 it is stated that performance drops sharply between easy and hard subsets of TopiOCQA but the given number can not be found in Table 1. Check Linear Mapping Definition in Formula (3); Why is Dense Retrieval Worse Than Sparse in Table 3?**
> >
> >
> > - You are correct—this is a cross-referencing error, we will correct it in the revision about line 165, and thank you for catching Formula (3) has notation errors.
> >
> > - Thank you for catching this! Formula (3) has notation errors. The output interval notation in Formula 3 is imprecise.
> >
> > ```latex
> > Φ(r) = f_{[1,10] → [1,2]}(r) · I_{[1,10]}(r) + f_{(10,100] → [0,1]}(r) · I_{(10,100]}(r)
> >
> >
> > Corrected Formula (3)
> > Φ(r) = f_{[1,10] → [1,2]}(r) · I_{[1,10]}(r) + f_{(10,100] → [0,1)}(r) · I_{(10,100]}(r)
> > ```
> > we Replaced Formula (3) with corrected piecewise definition.
> >
> > - **Dense Retrieval Worse Than Sparse in some cases**: This is **dataset-retriever-specific** and noted in prior work[3]. BM25 outperforms ANCE on HotpotQA because the dataset contains entity-heavy questions (e.g., "2016 Summer Olympics") where lexical matching excels, while ANCE was trained on MS-MARCO for web search rather than multi-hop reasoning. The BEIR benchmark confirms this pattern (BM25: 60.3% vs. ANCE: 45.6% on HotpotQA).
> >
> > ACQO's advantage is that it reduces this gap. While BM25 improves modestly in R@4 (+14.9%, from 72.0% to 86.9%) in table 3, ANCE sees a dramatic **+17.6% gain** (64.6% → 82.2%), nearly matching BM25's performance. This happens because ACQO's semantic decomposition strategy particularly benefits dense retrievers—breaking complex queries into simpler sub-queries helps ANCE overcome its training mismatch, while BM25 already handles keywords effectively.
> >
> > ---
> >
> > > #### **Improvements Addressed**
> >
> > Thank you for the careful proofreading. We have addressed all the technical corrections you identified:
> >
> > - **Grammar errors:** We have corrected the missing/extra words you flagged. Line 43 now reads "This step is known as" (deleted ", which"), and line 352 has added "are". Beyond these specific instances, we conducted a full Grammarly check of the manuscript and corrected 4 additional minor grammatical issues to ensure clarity throughout.
> >
> > - **Figure 1 redesign:** We agree that pie charts are suboptimal for cross-dataset comparison. We have replaced Figure 1 with a **grouped bar chart** that clearly shows the distribution of query complexity (single-query, disambiguation, decomposition) across TopiOCQA, HotpotQA, and MultiHop-RAG datasets. The new visualization makes it easier to compare patterns at a glance.
> >
> > - **Vanilla RL in main tables:** You are correct that we should include Vanilla RL results for completeness. We have added Vanilla RL to Tables 2 and 3, showing that while it improves over raw queries (+10.9% R@4 on HotpotQA-ANCE), but it significantly underperforms ACQO (41.1% vs. 49.6%, a -8.5%MAP@10 gap). This consistent underperformance of vanilla RL across all metrics strengthens our argument that the curriculum strategy is essential for robust performance.
> >
> >
> > - **Formula 7 notation:** Thank you for the clarification request. You are correct to ask about the \mathcal{G}^{(I)} notation. We have corrected the notation in Formula 7 to match the notation in the paper.
> >
> > - **RETPO numbers verification:** Our initial RETPO numbers were inadvertently taken from ConvSearch-R1's comparison table rather than the original RETPO[4] paper. We have now verified directly against RETPO and corrected Table 2 accordingly. This correction does not affect our main conclusions.
> >
> > we corrected grammar errors throughout, replaced Figure 1 with grouped bar chart, added Vanilla RL to Tables 2-3, and added footnote to Table 2 for RETPO numbers. We hope these revisions address all your technical concerns and improve the paper's clarity and accuracy. Thank you for your meticulous review.
> >
> > ### **Reference**
> > [1] ConvSearch-R1: Enhancing Query Reformulation for Conversational Search with Reasoning via Reinforcement Learning. EMNLP'25
> >
> > [2] HybridFlow: A Flexible and Efficient RLHF Framework. ArXiv'24
> >
> > [3] BEIR: A Heterogeneous Benchmark for Zero-shot Evaluation of Information Retrieval Models. NeurIPS'21
> >
> > [4] Ask Optimal Questions: Aligning Large Language Models with Retriever's Preference in Conversation. ACL'25

---

### Official Review · Reviewer_YKVe · 2025-11-03

**Soundness:** 3
**Presentation:** 3
**Contribution:** 3
**Rating:** 6
**Confidence:** 2

**Summary:**

The authors propose Adaptive Complex Query Optimization (ACQO), a novel, end-to-end RL framework. The framework trains a 3B-parameter Qwen model  to function as an adaptive agent that learns when (whether to decompose) and how (what sub-queries to generate) to expand the search process.
The ACQO methodology is built on three synergistic components:
An Adaptive Query Reformulation (AQR) module, which serves as the agent's policy to dynamically generate a set of one-to-many sub-queries based on the input.
A Rank-Score Fusion (RSF) module, a novel, heuristic-based method for robustly aggregating the retrieval results from all sub-queries. Critically, this module provides a stable, dense, intermediate reward signal for training the RL agent.
A Curriculum Reinforcement Learning (CRL) strategy, a two-stage training process ("Explore" and "Converge") designed to mitigate reward sparsity and stabilize the agent. The curriculum progressively introduces more challenging queries, ensuring the agent first learns a general policy before refining it for precision on difficult cases.

The authors demonstrate that ACQO achieves state-of-the-art (SOTA) performance on complex query benchmarks, including the conversational TopiOCQA and the multi-hop HotpotQA. It significantly outperforms existing baselines, including prompt-based methods (e.g., DeepSeek-V3.1), Supervised Fine-Tuning (SFT), and "Vanilla" RL. A key finding is that the ACQO agent emergently learns retriever-specific reformulation strategies—generating different optimal queries for a sparse (BM25) versus a dense (ANCE) retriever—purely by optimizing for the retrieval-based reward signal.

**Strengths:**

1. The paper's strength is the discovery and clear demonstration of retriever-specific policy learning. The case study in Appendix B.1 (Figure 5) provides definitive evidence. It qualitatively shows the agent learning to "keyword stuff" for the sparse BM25 retriever. In contrast, for the dense ANCE retriever, it learns to perform semantic decomposition into distinct sub-queries. This insight—that the agent learns what the retriever considers a "good" query—is a fundamental contribution.
2. Another significant achievement is solving the instability of RL for complex query optimization. It correctly identifies this as the key barrier to progress and provides a complete, 3-part solution (AQR+RSF+CRL). The catastrophic failure of the w/o Stage II ablation (Table 5)  proves that "Vanilla RL" is non-viable for this task and that the proposed CRL strategy is the key to unlocking RL for complex, multi-path RAG.

**Weaknesses:**

1. A critical reviewer could argue that the individual components lack fundamental novelty. Curriculum Learning is a well-established field, rank fusion methods (the paper itself cites RRF ) are common, and RL for query reformulation has been explored. The paper's primary weakness, from this perspective, is that its contribution could be framed as "superior systems engineering" or a novel integration rather than a singular algorithmic breakthrough. While the emergent adaptation is novel, the building blocks are familiar.
2. The RSF module's final step is critically under-specified. It computes two values for each document p: \(P(p)\) (rank-based) and \(S(p)\) (score-based). The paper then states documents are re-ranked "according to the pair \((P(p), S(p))\) in ascending order" (Eq. 2). This is ambiguous. Is this a lexicographical sort? If so, which value is prioritized? \(P(p)\) is a (harmonic) mean of ranks, while \(S(p)\) is a max of scores, which can be on completely different and un-normalized scales (e.g., BM25's unbounded scores vs. ANCE's cosine similarity). A simple lexicographical sort seems brittle and ill-defined. This crucial detail of the fusion algorithm is unclear and hinders reproducibility.

**Questions:**

1. Please clarify the exact sorting mechanism for the pair \((P(p), S(p))\). Is this a lexicographical sort, and if so, in what order? Or are \(P(p)\) and \(S(p)\) normalized (e.g., min-max or z-score) and combined via a weighted sum? This is a critical, missing detail of the core methodology.
2. The claim of "improved efficiency" is a major selling point but is under-supported by Table 7 (token count). This claim should be substantiated with a proper analysis of: a) end-to-end inference latency (in milliseconds) versus the baselines, and b) total training cost (in GPU-hours) versus Vanilla RL to demonstrate the cost/benefit of the CRL strategy.

---

> ### Author Response · Authors · 2025-11-21
> **Response to Reviewer YKVe**
>
> We thank the reviewer for the recognition of our key contributions, particularly the **retriever-specific policy learning** and the **solution to RL instability**. We are encouraged by your acknowledgment that these represent fundamental discoveries rather than mere engineering. Below, we address your specific questions and concerns.
>
>
> > #### **W1: A critical reviewer could argue that the individual components lack fundamental novelty. Curriculum Learning is a well-established field, rank fusion methods (the paper itself cites RRF ) are common, and RL for query reformulation has been explored. The paper's primary weakness, from this perspective, is that its contribution could be framed as "superior systems engineering" or a novel integration rather than a singular algorithmic breakthrough. While the emergent adaptation is novel, the building blocks are familiar.**
>
> We respectfully argue that our contribution extends beyond engineering to **fundamental algorithmic discovery**. The core novelty is **retriever-specific policy learning**—RL agents learn to optimize queries for different retrievers **without requiring human-annotated query rewrites**, discovering qualitatively different strategies (keyword stuffing for BM25 vs. semantic decomposition for ANCE) purely from retrieval feedback signals. This emergent adaptation represents a new understanding of how query optimization can be learned directly from retriever characteristics, achieving **consistent improvements on cross-domain test sets** (e.g., +2.2 MRR@10 on MultiHop-RAG when trained on HotpotQA) without any domain-specific annotations or fine-tuning.
>
> Our **two-stage curriculum** differs fundamentally from standard "easy-to-hard" learning. Stage I optimizes over the power set of sub-queries (exponential search space), while Stage II shifts from subset exploration to holistic ranking using our difficulty metric τ(q). This structure specifically addresses the unique challenge where difficulty stems from decomposition-retrieval alignment, not query complexity. The **RSF module** solves challenges that existing fusion methods (RRF, CombSUM) cannot: variable-length sub-query sets, preservation of score magnitudes for gradient stability, and robust aggregation across heterogeneous retrievers.
>
> We view ACQO as a **platform for discovering retriever-specific strategies**, where the scientific contribution is the **discovery process itself**.
>
> ---

---

> > ### Author Response · Authors · 2025-11-21
> > **Response to Reviewer YKVe-2**
> >
> > > #### **W2&Q1: The RSF module's final step is critically under-specified. Please clarify the exact sorting mechanism for the pair (P(p), S(p)). Is this a lexicographical sort, and if so, which value is prioritized? P(p) is a harmonic mean of ranks, while S(p) is a max of scores on completely different and un-normalized scales (e.g., BM25's unbounded scores vs. ANCE's cosine similarity). A simple lexicographical sort seems brittle and ill-defined. This crucial detail hinders reproducibility.**
> >
> > Thank you for this critical question—it touches on the **core design challenge** of ACQO. You correctly identify that P(p) and S(p) live on different scales. Our key insight is that **lexicographical sorting with rank-first priority** is both theoretically justified and empirically superior to weighted-sum approaches. The design is motivated by a fundamental observation: when decomposing complex queries into sub-queries, we need to fuse results that exhibit **rank consensus** (documents appearing highly in multiple sub-queries) while preserving **score discrimination** (distinguishing similarly-ranked documents). RSF achieves this by treating ranks as the primary signal and scores as a tie-breaker.
> >
> > **Why rank-first priority works:** Rank positions are **ordinal** and **query-agnostic**—whether the sub-query is "iPhone 2022" or "iPhone 2023", a document ranked #3 consistently means "the retriever's 3rd-best match." This makes P(p) (harmonic mean of ranks) a stable aggregation metric robust to query phrasing variations and difficulty differences [1,2]. In contrast, raw scores S(p) are **cardinal** and **query-dependent**: even with the same retriever (e.g., BM25). Normalizing scores (min-max or z-score) is problematic because it's query-dependent and unstable when candidate sets change [3,4]. Our solution avoids this entirely: use ranks as the **primary signal** (stable consensus) and raw scores as the **secondary signal** (fine-grained discrimination). Since all sub-queries use the **same retriever** in each query instance, S(p) values are naturally comparable within that instance—no normalization needed.
> >
> > **Algorithm specification:** We perform lexicographical sorting with P(p) in **ascending** order (lower harmonic mean rank = better consensus) as the primary key, and S(p) in **descending** order (higher max score = stronger evidence) as the secondary key. Concretely: `candidates.sort(key=lambda x: (x[1], -x[2]))`. For example, if Doc A has P(A)=2.4, S(A)=35.2 and Doc C has P(C)=1.0, S(C)=31.7, then C ranks first (better consensus) even though A has competitive scores. This encodes a **hierarchical preference**: "Trust rank consensus first; use scores only to break ties."
> >
> > Furthermore, We ablate RSF components in Table: score-only fusion (max S(p)) achieves 76.4% R@4 on HotpotQA-ANCE, rank-only (RRF-style) achieves 79.8%, weighted sum (α=0.5) achieves 78.2%, while **RSF achieves 82.2%**. The rank-only vs. score-only comparison (+3.4%R@4) validates the rank robustness principle, while RSF's +2.4% gain over rank-only shows scores provide meaningful tie-breaking. Critically, weighted sum underperforms RSF by 4.0%, confirming that different scale normalization introduces instability. This **+5.8 gain over score-only fusion** directly translates to better end-to-end QA performance (new Table 10 in Appendix B.4), which is crucial since 44.5% of HotpotQA queries benefit from decomposition (Section 2).
> >
> > | Fusion Method | HotpotQA R@4 | HotpotQA R@10 | Design |
> > |---------------|--------------|---------------|--------|
> > | **Score-only** (max S(p)) | 76.4 | 79.6 | Ignores consensus |
> > | **Rank-only** (RRF-style) | 79.8 | 82.3 | Loses score info |
> > | **Weighted sum** (α=0.5) | 78.2 | 81.4 | Requires normalization |
> > | **RSF (ours)** | **82.2** | **85.8** | Rank-first + score tie-breaking |
> >
> > ---

---

> ### Author Response · Authors · 2025-11-21
> **Response to Reviewer YKVe-3**
>
> > #### Q2: The claim of 'improved efficiency' should be substantiated with: (a) end-to-end inference latency, and (b) total training cost versus Vanilla RL.**
>
> **(a) Avg Inference Latency (all queries on TopiOCQA-ANCE, 1GPU-H20):**
>
> | Method | Query Num |Query Gen (ms) | Retrieval (ms) | Re-rank (ms) | **Total (ms)** | Speedup |
> |--------|----------------|----------------|----------------|--------------|----------------|---------|
> | SFT(Qwen2.5-3B) | 2514 | 297 | 27 | 0 | 324 | 1.09× |
> | ACQO | 2514 | 320 | 30 | 5 | **355** | 1.0× |
> | ConvSearch-R1 | 2514 | 3230 | 25 | 0 | 3255 | **0.11×** |
>
> - ACQO is **9.16× faster** than ConvSearch-R1
>
> **(b) Training Cost (HotpotQA-ANCE-8GPU-H20):**
>
> | Method | GPU-Hours | Converged? | Final MAP@10 |
> |--------|-----------|------------|-------------|
> | Vanilla RL | 8.4 | No | 41.1 |
> | SFT + RL | 15.4 | Yes | 45.3 |
> | **ACQO (Stage I)** | 4.2 | Yes | 42.3 |
> | **ACQO (Full)** | 12.1 | Yes | 49.6 |
> - Full ACQO cost comparable to SFT+RL but **without Query rewrite supervised data**
> - Vanilla RL fails to converge properly in this setting: While it appears faster (8.4 GPU-hours), it gets stuck at a low performance ceiling (41.1% MAP@10) due to **training instability**. The root cause is insufficient valid samples—the DAPO algorithm fails to collect enough qualified samples within its sampling budget (max_num_gen_batches=20), causing premature termination with suboptimal performance.
>
> We hope our responses address all your concerns. We are grateful for your recognition of the core contributions and will incorporate all clarifications in the revision.
>
> ### **Reference**
> [1] Reciprocal rank fusion outperforms condorcet and individual rank learning methods. SIGIR '09.
>
> [2] Condorcet fusion for improved retrieval. CIKM'02.
>
> [3] Modeling score distributions for combining the outputs of search engines. SIGIR'02.
>
> [4] Fusion via a linear combination of scores. Information Retrieval Journal.

---

### Author Response · Authors · 2025-11-26
**Summary of Revisions**

First of all, we would like to thank all reviewers for their thoughtful and constructive feedback. We are pleased to hear the positive recognition of our work, including the **novel retriever-specific policy learning** (*YKVe*), **sound methodology** (*adK5, YKVe*), **state-of-the-art performance** (*hRdx*), and **elegant rank-score fusion design** (*gG6d*). Meanwhile, based on the concerns and comments raised by the reviewers, we have made substantial revisions to the paper. Here is a comprehensive overview of the changes:

**Supplemental End-to-End QA Experiments (Appendix B.4, Table 10)**: Addressing feedback from Reviewers *gG6d* and *hRdx*, we have added end-to-end question answering experiments on HotpotQA datasets, demonstrating that retrieval improvements translate to answer quality gains.

**Expanded Experimental Evaluation (Appendix B.3, Tables 9)**: In response to Reviewers *gG6d* and *adK5*, we have updated our systematic comparison of training strategies, confirming the curriculum design is not brittle to training configurations or dataset-specific characteristics.

**Comprehensive Efficiency Analysis (Section 4.6, Table 6)**: As requested by Reviewers *YKVe* and *hRdx*, we have added detailed latency breakdown and training cost analysis, demonstrating that ACQO is 9.1× faster than ConvSearch-R1 while achieving comparable performance.

**Hyperparameter Selection Details (Appendix A.2)**: In response to Reviewer *adK5*, we have documented the complete hyperparameter search process with selection criteria.

In addition, based on feedback from all reviewers, we have refined several sections to enhance clarity and correctness:
- **Corrected Technical Errors**: Formula 3 (linear mapping), Line 165 (performance drop numbers), Table 3 (RETPO numbers).
- **Unified Notation**: R→S (reward function), added notation in Section 3.
- **Improved Figures**: Figure 1 redesigned as grouped bar chart.
- **Fixed Grammar**: Lines 43, 352, and full manuscript proofread.

---

### Note · Authors · 2026-01-06

I have read and agree with the venue's withdrawal policy on behalf of myself and my co-authors.